# HoloLLM: Multisensory Foundation Model for Language-Grounded Human Sensing and Reasoning

**Chuhao Zhou[1], Jianfei Yang[1]***

[1] MARS Lab, Nanyang Technological University

{chuhao002@e, jianfei.yang}@ntu.edu.sg

Project Page: https://ntumars.github.io/project/HoloLLM

Code: https://github.com/NTUMARS/HoloLLM

## Abstract

Embodied agents operating in smart homes must understand human behavior through diverse sensory inputs and communicate via natural language. While Vision-Language Models (VLMs) have enabled impressive language-grounded perception, their reliance on visual data limits robustness in real-world scenarios with occlusions, poor lighting, or privacy constraints. In this paper, we introduce HoloLLM, a Multimodal Large Language Model (MLLM) that integrates uncommon but powerful sensing modalities, such as LiDAR, infrared, mmWave radar, and WiFi, to enable seamless human perception and reasoning across heterogeneous environments. We address two key challenges: (1) the scarcity of aligned modality-text data for rare sensors, and (2) the heterogeneity of their physical signal representations. To overcome these, we design a Universal Modality-Injection Projector (UMIP) that enhances pre-aligned modality embeddings with fine-grained, text-aligned features from tailored encoders via coarse-to-fine cross-attention without introducing significant alignment overhead. We further introduce a human-VLM collaborative data curation pipeline to generate paired textual annotations for sensing datasets. Extensive experiments on two newly constructed benchmarks show that HoloLLM significantly outperforms existing MLLMs, improving language-grounded human sensing accuracy by up to 30%. This work establishes a new foundation for real-world, language-informed multisensory embodied intelligence.

## 1 Introduction

Embodied agents in smart homes, e.g., household robots and intelligent appliances, have garnered increasing attention in recent years [1–4]. To interact effectively with humans and execute real-world tasks, agents must understand human behavior and be capable of engaging in natural language communication. This necessitates the development of models that seamlessly integrate rich human perception with advanced language understanding and generation capabilities.

Vision-Language Models (VLMs) [5–8] have emerged as promising tools for enabling language-conditioned perception and reasoning. However, the visual modality alone struggles to operate in the real world, e.g., low-light environments, occlusions, and privacy-sensitive scenarios. In contrast, humans naturally rely on multiple sensory modalities, such as vision, audition, and olfaction, to perceive and adapt to diverse environments. Inspired by this biological principle, embodied agents can benefit from incorporating alternative sensing modalities, including LiDAR, infrared cameras, mmWave radar, and WiFi signals, to achieve more robust and comprehensive perception. Each of these modalities brings distinct advantages: LiDAR enables high-precision 3D reconstruction [9], infrared cameras support perception in darkness [10], and mmWave radar and WiFi are resilient to

---

*Corresponding Author (jianfei.yang@ntu.edu.sg)

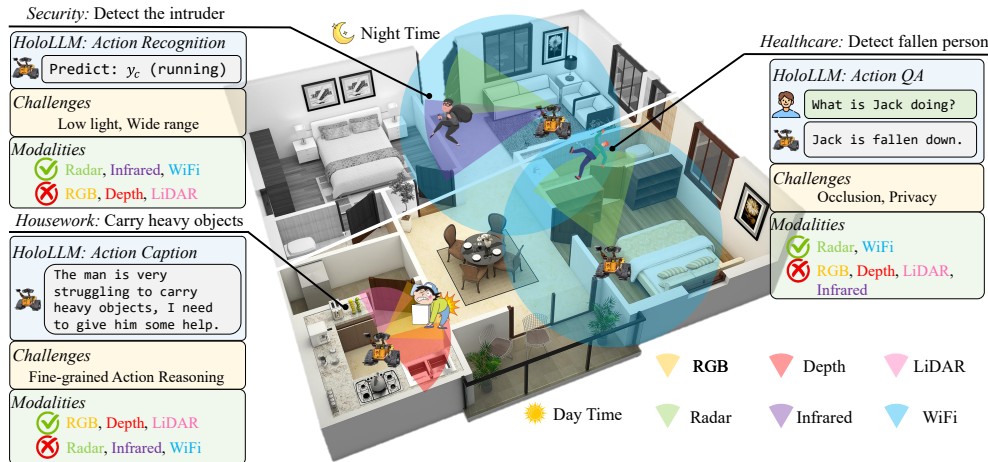

Figure 1: HoloLLM achieves *seamless* and *language-grounded* human perception and reasoning with complementary sensing modalities. It overcomes real-world challenges, e.g., illumination and privacy, with superior performance on human action recognition, question answering (QA), and captioning tasks, which enables embodied agents to work intelligently across diverse scenarios.

visual occlusions and lighting variations [11]. As illustrated in Fig. 1, vision alone fails to detect a fallen individual behind an obstruction, while radar- and WiFi-based modalities remain effective. Hence, we believe Multimodal Large Language Models (MLLMs) that integrate diverse sensor inputs can provide excellent adaptability and reliability in complex, real-world environments.

However, integrating sensing modalities into an MLLM is non-trivial due to two key challenges. First, we must align sensing modalities with text using limited training data. For common modalities such as RGB or depth images, millions of web-sourced 'modality-text' data pairs are available [12–14], which enable large-scale pre-training of multimodal projectors to effectively align common modalities with text. In contrast, other sensing modalities (e.g., mmWave and WiFi signals) are not available online, with only a few thousand samples collected in labs [11, 10]. The data scarcity complicates the alignment between sensing modalities and text. Secondly, modality-text alignment requires a robust feature encoder, but it is challenging to learn heterogeneous characteristics of these rare sensing modalities. To model the physical world, different sensors are designed to leverage distinct physical designs (e.g., wavelengths and frequencies) at multiple granularities, of which the sensing data show extreme heterogeneity and significantly differ from common modalities, leading to huge challenges for existing transformer-based encoders to learn robust representations [10, 15]. These two challenges constitute the question investigated in this work: **can we enable MLLM to learn data-scarce and heterogeneous sensing modalities for language-grounded human perception and reasoning?**

To this end, our key idea comes from two aspects. Due to data scarcity, directly aligning sensing modalities with text through large-scale pre-training is infeasible. To address it, we propose to generate initial embeddings for each modality pre-aligned with text, without additional training. Then, only minor data and fine-tuning are sufficient to achieve appropriate alignment. Regarding the heterogeneity of sensing modalities, previous studies [16, 17] show that tailored encoders are more effective than normal transformers to extract modality-specific features. However, directly aligning raw modality-specific features with text from scratch demands massive training data and incurs additional alignment overhead. As a result, we take raw features as references and integrate text-aligned features into multimodal embeddings via an iterative process: querying and fusing text-aligned features into the multimodal embeddings through cross-attention, and projecting the embeddings into the LLM semantic space to form the enhanced queries for the next iteration.

In this paper, we propose HoloLLM, an MLLM for seamless human perception and reasoning across common and sensing modalities. Specifically, we adopt a CLIP vision encoder aligned with the text modality to generate pre-aligned initial embeddings for each modality. Tailored encoders are then designed to fully explore fine-grained, modality-specific discriminative features. To avoid additional alignment overhead, we propose the Universal Modality-Injection Projector (UMIP) for progressive modality features integration. UMIP takes the initial embeddings as coarse queries to adaptively

identify and integrate fine-grained, text-aligned modality features via coarse-to-fine cross-attention. Furthermore, we introduce a human-VLM collaborative data curation pipeline to generate textual annotations for existing multimodal human sensing datasets: MM-Fi [11] and XRF55 [10]. A comprehensive evaluation of state-of-the-art MLLMs is conducted, establishing the first benchmark for multimodal human perception and reasoning grounded in sensing modalities.

In summary, our contributions are threefold:

- We propose HoloLLM, the first work to align MLLM with rare sensing modalities to achieve seamless human perception and reasoning with language grounded.
- We propose Universal Modality-Injection Projector (UMIP) and modality-specific encoders to deal with the data-scarce feature learning and modality-text alignment, respectively.
- To evaluate HoloLLM, we design a human-VLM collaborative data curation pipeline to construct the text-paired multimodal dataset. Then, the first multisensory benchmark for human sensing is established with different settings and baselines. The HoloLLM shows superior performance, improving existing MLLM by around 30% on some QA tasks.

## 2 Related Work

**Multimodal Human Sensing.** Human sensing aims to perceive, analyze, and understand human actions, which is essential in human-agent interaction. Beginning with human sensing based on RGB and depth frames [18], various other sensing modalities, such as LiDAR [19], mmWave [20, 21], WiFi-CSI [17, 15], and RFID [10], have been progressively introduced to address limitations, including lighting variations, occlusions, and privacy concerns. To achieve more comprehensive human perception, methods that explore complementary information across modalities become dominant [22, 23, 11, 16]. With the development of LLM, many works seek to leverage the high-level semantics and strong generalization capability of language to perform zero-shot human sensing tasks [24], which are extended to multimodal inputs for more comprehensive recognition [25]. However, existing models cannot reason and generate responses based on perceived information, limiting their capacity to engage in language-based interaction with humans.

**Multimodal-Text Alignment.** Extending LLMs to other modalities to form MLLMs enables them to follow human instructions based on multimodal inputs such as images [5, 8], videos [26], audio [27], and point clouds [28]. Aligning multimodal inputs with text is the key challenge for MLLMs. Existing methods primarily rely on building multimodal projectors. Typically, Linear / MLP projectors [5, 26] directly project the multimodal inputs into the LLM text space. Despite the simplicity, the number of multimodal tokens will increase significantly when high-resolution inputs are presented. To alleviate this issue, Resampler / Q-Former projectors [8, 27] utilize a fixed number of learnable queries to align the most task-relevant multimodal information with the text. Recently, hybrid projectors have been developed [6, 7], which serve as trade-offs between MLP and Q-Former projectors, to fully preserve all information within multimodal inputs while minimizing the number of tokens generated. Nevertheless, these projectors commonly need large-scale 'modality-text' data pairs for pre-training, which are unavailable for sensing modalities.

## 3 Method

In this part, we first formulate our problem (Sec 3.1), then delve into the details of the UMIP (Sec 3.2). Followed by an introduction of the two-stage training strategy for HoloLLM (Sec 3.3). Finally, we present the data curation pipeline for our multisensory benchmarking (Sec 3.4).

### 3.1 Problem Formulation

The objective of MLLMs is to obtain multimodal tokens $\mathbf{Z}^m$, which are not only aligned with text tokens $\mathbf{Z}^{text}$ but also preserve modality-specific features within the inputs $\mathbf{X}^m$. This work aims to learn an MLLM with data-scarce and heterogeneous sensing modalities, which presents a greater challenge, as large-scale pre-training of multimodal projectors is infeasible.

An MLLM typically consists of three components: (1) A multimodal Encoder $E(\cdot)$ that converts the inputs $\mathbf{X}^m$ from modality $m$ into a sequence of multimodal embeddings $\mathbf{Y}^m$. (2) A projector $P(\cdot)$

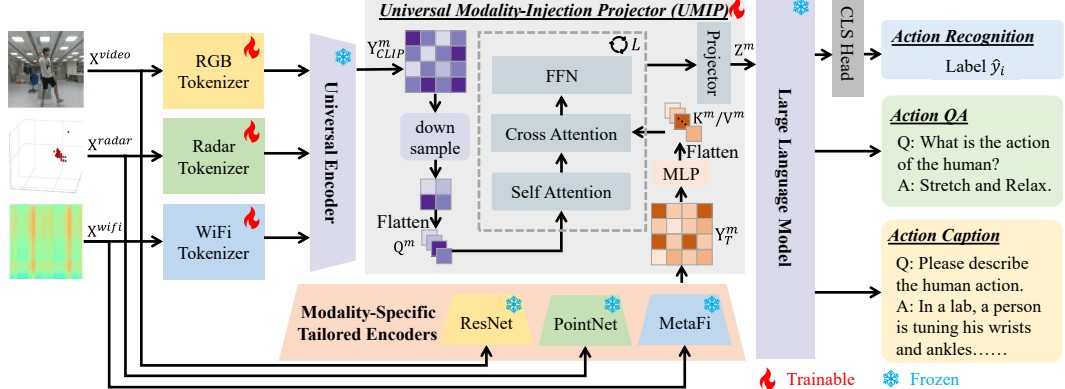

Figure 2: Architecture of HoloLLM. Given multimodal inputs $\mathbf{X}^m$, HoloLLM utilizes modality-specific tokenizers and a universal encoder to extract pre-aligned initial embeddings $\mathbf{Y}^m_{CLIP}$. Meanwhile, pre-trained tailored encoders are applied to explore modality features $\mathbf{Y}^m_T$. The UMIP then transforms $\mathbf{Y}^m_{CLIP}$ and $\mathbf{Y}^m_T$ into coarse queries $\mathbf{Q}^m$ and fine-grained keys and values $\mathbf{K}^m/\mathbf{V}^m$. By iteratively enhancing the queries via coarse-to-fine cross-attention and projecting them to the LLM text space, the aligned multimodal tokens $\mathbf{Z}^m$ fully enriched by modality features can be achieved.

to align $\mathbf{Y}^m$ with the LLM text space, forming aligned multimodal tokens $\mathbf{Z}^m = P(\mathbf{Y}^m)$. (3) An LLM $LLM(\cdot, \cdot)$, which takes $\mathbf{Z}^m$ and the text tokens $\mathbf{Z}^{text}$ corresponding to human instructions as inputs, to output the responses $\mathbf{A} = \{a_i\}_{i=1}^S = LLM(\mathbf{Z}^m, \mathbf{Z}^{text})$ in an auto-regressive approach: $p(\mathbf{A}) = \prod_{i=1}^S p(a_i \mid \mathbf{Z}^m, \mathbf{Z}^{text}, \mathbf{A}_{<i})$, $S$ is the length of responses generated by $LLM$.

## 3.2 Universal Modality-Injection Projector

As shown in Fig. 2, we propose an efficient projector, named the Universal Modality-Injection Projector (UMIP). To overcome the data scarcity, we attempt to generate initial embeddings pre-aligned with the text for each modality, without the need for extra training. Specifically, we take the CLIP vision encoder [29] as a unified multimodal encoder $E_{CLIP}(\cdot)$ to obtain initial embeddings $\mathbf{Y}^m_{CLIP}$ for modality $m$:

$$\mathbf{Y}^m_{CLIP} = E_{CLIP}(\mathbf{X}^m) \in \mathbb{R}^{n_m \times d_m}, \tag{1}$$

where $n_m$ and $d_m$ denote the number and dimension of the embeddings. Benefiting from extensive image-text contrastive pre-training, the CLIP vision encoder achieves superior alignment with text and offers transferability to other modalities [30]. Consequently, the $\mathbf{Y}^m_{CLIP}$ can be considered inherently pre-aligned with text, and minor data and fine-tuning are sufficient to align them with text.

However, our experiments (Tab. 1 and Tab. 2) reveal that the initial embeddings $\mathbf{Y}^m_{CLIP}$ lack sufficient discriminability. Previous studies [16, 17, 10] show that dedicated convolutional encoders outperform transformer counterparts in capturing heterogeneous spatial features from sensing modalities grounded in radio frequency, such as WiFi signals. To this end, we design a convolutional tailored encoder $E^m_T(\cdot)$ for each modality to capture the heterogeneous modality features $\mathbf{Y}^m_T$:

$$\mathbf{Y}^m_T = \text{MLP}^m(E^m_T(\mathbf{X}^m)) \in \mathbb{R}^{h_m \times w_m \times d_m}, \tag{2}$$

where $\text{MLP}^m(\cdot)$ is a MLP projector to align the feature dimension of $E^m_T(\cdot)$ to that of CLIP $d_m$, $h_m$ and $w_m$ are the height and width of the feature map.

The significant heterogeneity of $\mathbf{Y}^m_T$ makes directly aligning them with the text inefficient, increasing the demand for training data. Therefore, we only take $\mathbf{Y}^m_T$ as references to provide fine-grained, modality-specific features and convert them into candidate keys and values: $\mathbf{K}^m, \mathbf{V}^m \in \mathbb{R}^{h_m w_m \times d_m}$. Meanwhile, the pre-aligned initial embeddings $\mathbf{Y}^m_{CLIP}$ are downsampled to form the queries:

$$\mathbf{Q}^m = \text{AvgPool}(\mathbf{Y}^m_{CLIP}) \in \mathbb{R}^{n'_m \times d_m}, \tag{3}$$

where AvgPool is 1D adaptive average pooling and $n'_m < n_m$ limits the number of queries to a fixed value. The queries $\mathbf{Q}^m$ only contain the coarse-grained modality features, which are enhanced via:

$$
\begin{aligned}
\mathbf{Q}^m_{(l)} &= \text{SelfAtt}(\mathbf{Q}^m_{(l-1)}), \\
\mathbf{Q}^m_{(l)} &= \text{CrossAtt}(\mathbf{Q}^m_{(l)}, \mathbf{K}^m, \mathbf{V}^m), \\
\mathbf{Q}^m_{(l)} &= \text{FFN}(\mathbf{Q}^m_{(l)}), \quad l = 1, \dots, L,
\end{aligned}
\tag{4}
$$

where $\mathbf{Q}^m_{(0)} = \mathbf{Q}^m$. SelfAtt($\cdot$), CrossAtt($\cdot, \cdot, \cdot$), and FFN($\cdot$) are self-attention, cross-attention, and feedforward layers in each block of UMIP, respectively, and $L$ is the number of blocks.

Our UMIP follows an iterative process to produce discriminative multimodal tokens that are adequately aligned with the text. In each block of UMIP, the coarse-to-fine cross-attention adaptively identifies the text-aligned modality features from $\mathbf{V}^m$ and injects them into the queries $\mathbf{Q}^m$ for enhancement. The updated queries are projected into the text space via the feedforward layer, which serves as enhanced queries for the next block. Finally, the multimodal tokens from the last block $\mathbf{Z}^m = \text{MLP}(\mathbf{Q}^m_L)$ can be aligned with the LLM text space while sufficiently enriched by modality-specific features. Here, MLP($\cdot$) maps the dimension of CLIP (1024) to that of the LLM (4096).

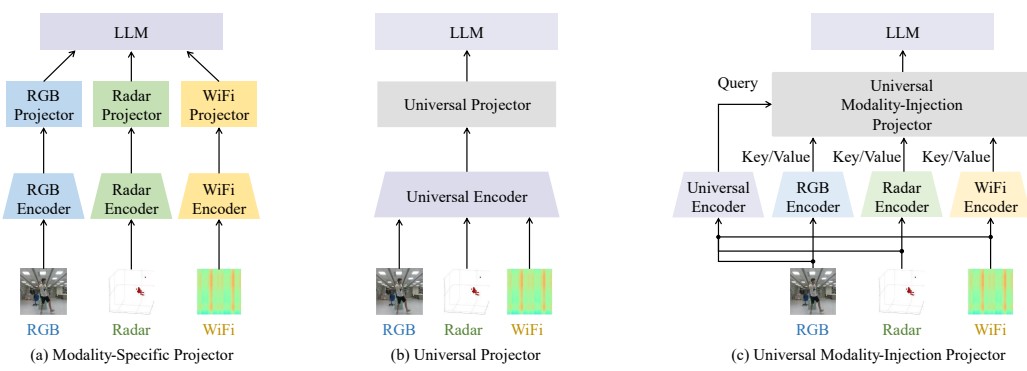

Figure 3: Comparison between UMIP and other projectors: (a) Modality-Specific Projector [28, 27, 31, 32], (b) Universal Projector [30], and (c) Universal Modality-Injection Projector (Ours).

**Discussion on prior art.** We compare UMIP with state-of-the-art multimodal projectors in Fig. 3. Specifically, most existing works [28, 27, 31, 32] adopt modality-specific encoders and projectors (Fig. 3 (a)), which commonly requires substantial 'modality-text' data pairs for pre-training. As shown in Fig. 3 (b), OneLLM [30] attempts to handle various modalities via a unified framework that consists of a universal encoder and projector. However, without a dedicated design for capturing heterogeneous spatial features, the universal encoder struggles to obtain sufficiently discriminative multimodal tokens. Different from existing works, UMIP only utilizes the universal encoder to generate initial embeddings for each modality (Fig. 3 (c)). These embeddings are then progressively enhanced by fine-grained, text-aligned features from tailored encoders.

## 3.3 Training Strategy and Objectives

To effectively train HoloLLM, we propose a two-stage training strategy: (1) pre-training tailored encoders to extract modality-specific features, and (2) fine-tuning the HoloLLM to learn discriminative multimodal tokens that are appropriately aligned with the text space.

During the first stage, task-specific objective, i.e., the classification loss for the Human Action Recognition (HAR) task, is utilized to pre-train tailored encoders:

$$
L_1 = CE(\text{Classifier}(E^m_T(\mathbf{X}^m_i)), c_i),
\tag{5}
$$

where $E^m_T$ is the tailored encoder for modality $m$, $\mathbf{X}^m_i$ is the $i$-th sample in the dataset, and $c_i$ is the corresponding action label. $CE(\cdot, \cdot)$ and Classifier($\cdot$) refer to the cross-entropy loss and the classifier to predict the action label, respectively.

After pre-training, all tailored encoders are frozen. The modality-specific tokenizers and the UMIP are then fine-tuned by the combination of task-specific and next-token prediction objectives:

$$L_2 = CE(\text{Classifier}(LLM(\mathbf{Z}_i^m, \mathbf{Z}_i^{text}), c_i) + L_{next}, \tag{6}$$

where $\mathbf{Z}_i^m$ and $\mathbf{Z}_i^{text}$ denote the multimodal tokens and the instruction text tokens for the $i$-th sample, $L_{next}$ is the cross-entropy loss of next-token prediction for both Action QA and Action Caption tasks.

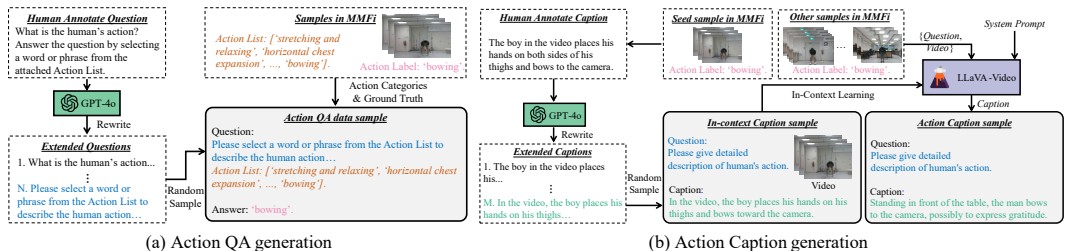

Figure 4: Data curation pipeline for (a) Action question answering (QA) and (b) Action Caption.

## 3.4 Data Curation for Multisensory Language-grounded Benchmarking

To perform real-world tasks in diverse smart home scenarios, embodied agents must understand human behavior from multisensory inputs and engage in language-grounded communication. Therefore, we establish the first multisensory benchmark for evaluating human perception and reasoning capabilities in HoloLLM, encompassing action recognition, question answering (QA), and captioning tasks. However, textual descriptions for sensing modalities are not available in public datasets and online sources. Inspired by other modalities such as point clouds [33] and IMU [34], we select two multimodal human sensing datasets, MM-Fi [11] and XRF55 [10], and refer to the vision modality to generate action QA and caption data. As shown in Fig. 4, we take the MM-Fi dataset as an example to illustrate our data curation pipeline; more details can be found in Appendix A.

**Action QA generation.** In this work, Action QA is conditioned on options. To ensure precision and diversity, 5 questions are first annotated by human experts and rewritten by GPT-4o [35], resulting in a list of 15 questions. For a sample from each modality, a question ('*Question*') is randomly sampled from the list. Moreover, all action categories of the MM-Fi dataset, along with the action label of the data sample, are appended to the question to generate the '*Action List*' (options) and the '*Answer*'. Formally, an Action QA sample can be formulated as: {*Question*, *Action List*, *Answer*}.

**Action Caption generation.** We propose a human-VLM collaborative pipeline to generate captions for sensing modalities. Specifically, a fixed question ('*Question*': '*Please give detailed descriptions of human's action.*') is designed for all caption tasks. Then, we uniformly select a small set of samples across various environments, subjects, and action categories to form the 'In-context Caption Samples'. For other data samples, LLaVA-Video [26] is adopted to automatically generate the captions ('*Caption*') via an in-context learning manner. Finally, an Action Caption sample can be formulated as: {*Question*, *Caption*}, which is shared among all modalities of a data sample.

## 4 Experiment

### 4.1 Experimental Settings

**Datasets.** We utilize two multimodal human-sensing datasets **MM-Fi** [11] and **XRF55** [10] with generated textual descriptions. Specifically, MM-Fi consists of 5 modalities: Video (V), Depth images (D), LiDAR (L), mmWave Radar (M), and WiFi-CSI (W). Besides, XRF55 also contains 5 modalities: Video (V), Depth images (D), Infrared images (I), RFID signals (R), and WiFi-CSI (W).

**Benchmarks.** To comprehensively evaluate various MLLMs across diverse scenarios, we design three experimental settings: (1) Random Split (Random), (2) Cross-Subject Split (CrossSub), and (3) Cross-Environment Split (CrossEnv). Specifically, 'Random' involves a random split of all samples

with a ratio of 3:1, and 'CrossSub' / 'CrossEnv' selects samples from nonoverlapping human subjects / environments for training and testing. For quantitative evaluation, we use the accuracy for Action Recognition and Action QA, and the METEOR [36] metric for Action Caption.

**Implementation Details.** Following OneLLM [30], we take CLIP VIT Large pre-trained on LAION [12] to provide initial multimodal embeddings, and the LLaMA2-7B [37] as our LLM. For tailored encoders, we take Resnet18 [38] for Vision, Depth and Infrared, PointNet [39] for LiDAR and mmWave, 1D Temporal Resnet18 [10] for RFID, and MetaFi [17] for WiFi modalities. The UMIP contains $L = 8$ blocks, with 64 query tokens for Vision, Depth, mmWave, and Infrared, 256 query tokens for LiDAR and WiFi (XRF55), and 16 query tokens for RFID and WiFi (MM-Fi). Please refer to Appendix B for more details on our experimental settings.

| Settings | Models | Sources | Types | Human Action QA (Accuracy) | | | | | | Action Caption (METEOR) | | | | | |
|---|---|---|---|---|---|---|---|---|---|---|---|---|---|---|---|
| | | | | V | D | M | L | W | Avg | V | D | M | L | W | Avg |
| Random | Tokenpacker | arXiv'24 | Proj | 2.7 | 2.2 | 2.6 | 2.6 | 1.9 | 2.4 | 8.9 | 8.9 | 8.9 | 8.9 | 8.9 | 8.9 |
| | Honeybee | CVPR'24 | Proj | 2.3 | 2.5 | 2.0 | 1.7 | 2.0 | 2.1 | 10.0 | 10.0 | 10.1 | 10.3 | 10.2 | 10.1 |
| | OneLLM | CVPR'24 | Proj | 3.6 | 3.5 | 4.7 | 3.5 | 4.2 | 3.9 | 15.5 | 15.3 | 13.3 | 15.9 | 16.6 | 15.3 |
| | ImageBind | CVPR'23 | Enc | 89.3 | 76.7 | 45.8 | 11.1 | 8.0 | 46.2 | 28.4 | 21.2 | 18.8 | 16.2 | 14.8 | 19.9 |
| | HoloLLM | - | Proj | **99.8** | **99.7** | **95.8** | **84.2** | **52.8** | **86.5** | **30.8** | **31.1** | **29.6** | **27.4** | **23.0** | **28.4** |
| CrossSub | Tokenpacker | arXiv'24 | Proj | 3.0 | 3.3 | 3.6 | 3.2 | 3.5 | 3.3 | 6.6 | 6.5 | 6.6 | 6.6 | 6.4 | 6.5 |
| | Honeybee | CVPR'24 | Proj | 1.8 | 1.8 | 1.9 | 2.2 | 2.0 | 1.9 | 10.3 | 10.3 | 10.3 | 10.3 | 10.2 | 10.3 |
| | OneLLM | CVPR'24 | Proj | 3.8 | 3.4 | 4.0 | 2.6 | 4.6 | 3.7 | 15.2 | 14.6 | 10.3 | 15.0 | 16.1 | 14.2 |
| | ImageBind | CVPR'23 | Enc | 76.9 | 43.3 | 45.5 | 6.8 | 7.7 | 36.0 | 25.8 | 21.0 | 20.9 | 15.3 | 16.5 | 19.9 |
| | HoloLLM | - | Proj | **98.0** | **98.9** | **88.0** | **66.5** | **8.0** | **71.9** | **30.6** | **30.5** | **29.5** | **24.9** | **16.7** | **26.4** |
| CrossEnv | Tokenpacker | arXiv'24 | Proj | 5.0 | 5.0 | 4.7 | 4.2 | 4.3 | 4.6 | 3.9 | 3.9 | 3.8 | 3.8 | 3.7 | 3.8 |
| | Honeybee | CVPR'24 | Proj | 2.0 | 1.5 | 1.5 | 1.9 | 1.8 | 1.7 | 10.4 | 10.5 | 10.4 | 10.6 | 10.4 | 10.4 |
| | OneLLM | CVPR'24 | Proj | 4.2 | 8.0 | 1.1 | 7.8 | 4.0 | 5.0 | 15.5 | 13.1 | 2.2 | 5.1 | 10.4 | 9.3 |
| | ImageBind | CVPR'23 | Enc | 41.0 | 5.3 | 24.0 | 7.6 | 5.5 | 16.7 | 19.4 | 19.8 | 17.5 | 15.0 | 14.9 | 17.3 |
| | HoloLLM | - | Proj | **79.5** | **91.6** | **61.4** | **41.4** | **8.2** | **56.4** | **25.7** | **27.5** | **24.5** | **19.6** | **15.9** | **22.6** |

Table 1: Evaluation of Human Action QA and Caption tasks on MM-Fi [11] across three settings. The Accuracy (%) and METEOR (%) are adopted for Action QA and Caption, respectively

| Settings | Models | Sources | Types | Human Action QA (Accuracy) | | | | | | Action Caption (METEOR) | | | | | |
|---|---|---|---|---|---|---|---|---|---|---|---|---|---|---|---|
| | | | | V | D | I | R | W | Avg | V | D | I | R | W | Avg |
| Random | Tokenpacker | arXiv'24 | Proj | 1.2 | 1.2 | 1.4 | 1.1 | 1.2 | 1.2 | 8.0 | 7.9 | 7.7 | 7.9 | 7.7 | 7.8 |
| | Honeybee | CVPR'24 | Proj | 1.4 | 1.6 | 1.5 | 1.6 | 1.5 | 1.5 | 10.0 | 10.1 | 10.3 | 10.1 | 10.0 | 10.1 |
| | OneLLM | CVPR'24 | Proj | 2.0 | 1.9 | 1.5 | 1.9 | 2.3 | 1.8 | 14.5 | 13.1 | 13.7 | 11.6 | 13.7 | 13.3 |
| | ImageBind | CVPR'23 | Enc | 62.2 | 22.2 | 79.0 | 5.3 | 10.0 | 35.8 | 19.3 | 13.0 | 24.3 | 12.3 | 12.7 | 16.3 |
| | HoloLLM | - | Proj | **94.5** | **92.3** | **92.6** | **27.1** | **11.2** | **63.5** | **34.2** | **34.8** | **34.7** | **15.5** | **14.0** | **26.6** |
| CrossSub | Tokenpacker | arXiv'24 | Proj | 1.2 | 1.1 | 1.4 | 1.2 | 1.2 | 1.2 | 9.2 | 9.1 | 9.1 | 9.2 | 9.3 | 9.2 |
| | Honeybee | CVPR'24 | Proj | 1.3 | 1.3 | 1.4 | 1.5 | 1.3 | 1.4 | 9.3 | 9.3 | 9.3 | 9.4 | 9.3 | 9.3 |
| | OneLLM | CVPR'24 | Proj | 1.9 | 1.5 | 2.0 | 2.3 | 2.1 | 2.0 | 13.4 | 13.5 | 14.5 | 9.2 | 13.1 | 12.7 |
| | ImageBind | CVPR'23 | Enc | 13.3 | 11.1 | 20.4 | **3.8** | **4.9** | 10.7 | 15.8 | 14.6 | 18.3 | **12.7** | **14.5** | 15.2 |
| | HoloLLM | - | Proj | **44.3** | **42.1** | **38.3** | 3.4 | 3.6 | **26.3** | **22.3** | **23.1** | **22.8** | 11.8 | 13.7 | **18.7** |
| CrossEnv | Tokenpacker | arXiv'24 | Proj | 1.7 | 1.4 | 1.6 | 1.4 | 1.5 | 1.5 | 7.2 | 7.2 | 7.2 | 7.1 | 7.1 | 7.2 |
| | Honeybee | CVPR'24 | Proj | 1.5 | 1.4 | 1.6 | 1.2 | 1.1 | 1.3 | 11.2 | 11.2 | 11.0 | 11.0 | 11.3 | 11.1 |
| | OneLLM | CVPR'24 | Proj | 1.4 | 3.2 | 1.8 | 1.7 | 2.3 | 2.1 | 16.9 | 6.7 | 11.5 | 8.4 | 13.0 | 11.3 |
| | ImageBind | CVPR'23 | Enc | 4.7 | 4.9 | 16.9 | **2.8** | 2.6 | 6.4 | 13.1 | 14.3 | 16.9 | **12.7** | 12.0 | 13.8 |
| | HoloLLM | - | Proj | **25.9** | **8.6** | **22.1** | 2.6 | **2.7** | **12.4** | **19.8** | **14.7** | **17.1** | 10.8 | **13.7** | **15.2** |

Table 2: Evaluation of Human Action QA and Caption tasks on XRF55 [10] across three settings. The Accuracy (%) and METEOR (%) are adopted for Action QA and Caption, respectively

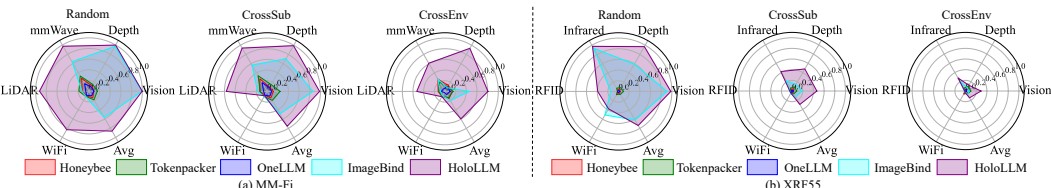

Figure 5: Evaluation of Human Action Recognition on MM-Fi [17] and XRF55 [10] across three benchmarks in terms of Accuracy (Better to zoom in).

## 4.2 Main Results

We compare HoloLLM with state-of-the-art MLLMs across three tasks: Action QA, Action Caption, and Action Recognition. Specifically, we divided existing methods into Encoder-based (Image-Bind [31]) and Projector-based (Honeybee [6], Tokenpacker [7], OneLLM [30]) methods. The Encoder-based methods aim to align multimodal embeddings from various encoders via contrastive learning, while the Projector-based methods focus on designing effective projectors for alignment. For a fair comparison, all methods are fine-tuned on MM-Fi and XRF55 datasets as our HoloLLM.

**Action QA and Action Caption.** The results on MM-Fi and XRF55 datasets are summarized in Tab. 1 and Tab. 2. For all three benchmarks, HoloLLM outperforms other MLLMs by a large margin on almost all modalities (indicated in **bold**). Specifically, Tokenpacker and Honeybee only use a modality-shared projector, which cannot effectively capture modality-specific features aligned with text. By introducing learnable queries for each modality to the projector, OneLLM and ImageBind can better align multimodal embeddings with the text space while exploring the modality features. Moreover, ImageBind employs dedicated encoders for depth and infrared modalities, along with a stronger vision encoder (CLIP ViT-H), enabling more effective capture of modality-specific features and improved performance. Instead of using simple learnable queries, HoloLLM designs tailored encoders to adequately capture fine-grained, modality-specific features, which are adaptively injected into the aligned multimodal tokens via UMIP. The results demonstrate that multimodal alignment and discriminability are equally critical for human perception and reasoning based on sensing modalities.

**Action Recognition.** As shown in Fig. 5, HoloLLM outperforms other MLLMs on Action Recognition compared across most modalities. For some sensing modalities, such as WiFi and RFID, performances on par with other MLLMs are observed under "CrossSub" or "CrossEnv" settings. Some sensing modalities are highly sensitive to different subjects or environments, making it challenging to achieve cross-subject or cross-environment generalization. More efforts should be made toward building large-scale sensing datasets to further promote their generalization capability.

| MMFi | Action Recognition | | | | | | Action QA | | | | | | Action Caption | | | | | |
|---|---|---|---|---|---|---|---|---|---|---|---|---|---|---|---|---|---|---|
| | V | D | M | L | W | Avg | V | D | M | L | W | Avg | V | D | M | L | W | Avg |
| Baseline | 9.7 | 9.5 | 18.6 | 6.5 | 7.4 | 10.3 | 3.9 | 3.3 | 14.4 | 4.8 | 4.7 | 6.2 | 15.3 | 15.1 | 17.4 | 15.9 | 16.0 | 15.9 |
| +TailorEncoder | 83.1 | 91.8 | 58.2 | 50.4 | 8.8 | 58.5 | 71.8 | 73.4 | 48.4 | 28.0 | 11.3 | 46.6 | 25.9 | 25.4 | 24.0 | 20.3 | 13.8 | 21.9 |
| +UMIP | 80.6 | 93.2 | 61.0 | 53.0 | 9.6 | 59.5 | 79.5 | 91.6 | 61.4 | 41.4 | 8.2 | 56.4 | 25.7 | 27.5 | 24.5 | 19.6 | 15.9 | 22.6 |
| XRF55 | V | D | I | R | W | Avg | V | D | I | R | W | Avg | V | D | I | R | W | Avg |
| Baseline | 4.4 | 3.8 | 18.2 | 2.6 | 3.4 | 6.5 | 5.7 | 2.5 | 2.4 | 3.2 | 3.8 | 3.5 | 12.0 | 12.1 | 13.0 | 11.1 | 13.5 | 12.3 |
| +TailorEncoder | 27.5 | 9.1 | 31.2 | 1.7 | 2.5 | 14.4 | 23.5 | 8.0 | 23.0 | 1.7 | 2.7 | 11.8 | 20.2 | 11.7 | 20.1 | 10.4 | 12.3 | 14.9 |
| +UMIP | 28.9 | 12.4 | 28.3 | 1.7 | 3.7 | 15.0 | 25.9 | 8.6 | 22.1 | 2.6 | 2.7 | 12.4 | 19.8 | 14.7 | 17.1 | 10.8 | 13.7 | 15.2 |

Table 3: Ablations results. We conduct experiments on both MM-Fi and XRF55 under "Cross-Environment" setting to show the contribution of the key components.

## 4.3 Ablation Results

We conduct the ablation study on both MM-Fi and XRF55 datasets under the "CrossEnv" setting to show the effectiveness and generalization capability of key components. Specifically, 'Baseline' only adopts the CLIP ViT-L to extract multimodal embeddings and utilizes a Q-former [8] with modality-specific learnable queries as the projector. Following OneLLM, the size of the learnable queries is $\mathbb{R}^{30 \times 1024}$. The results are summarized in Tab. 3.

**Ablation Study on Tailored Encoders.** To evaluate the effectiveness of tailored encoders, we replace the CLIP ViT-L in the 'Baseline' with them and use the same Q-former for multimodal alignment ('+TailoredEncoder'). As shown in Tab. 3, introducing tailored encoders significantly improves performance across all modalities on both datasets. This demonstrates that fine-grained, modality-specific discriminative features are crucial in human perception and reasoning tasks, and tailored encoders can capture the features more effectively than the modality-shared unified encoder.

**Ablation Study on UMIP.** Furthermore, applying UMIP ('+UMIP') leads to performance improvement, especially for Action QA, which requires a deeper understanding of language-based human instructions and action categories. The results indicate that the multimodal tokens generated by UMIP achieve better alignment with text, thereby enhancing human action reasoning. Moreover, by

adaptively enhancing the pre-aligned initial embeddings using fine-grained, text-aligned modality features, UMIP can provide multimodal tokens with stronger discriminability.

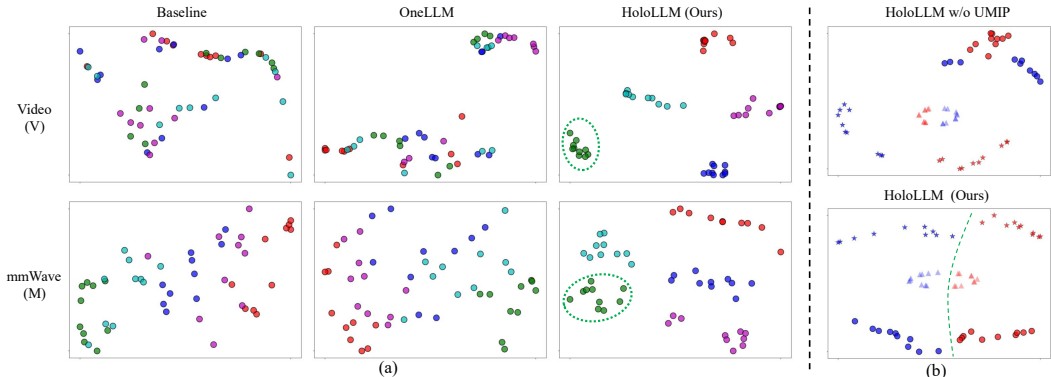

Figure 6: Visualization results by tSNE [40]. (a) Visualization of aligned tokens from 5 action categories (denoted by different colors) generated by Baseline, OneLLM [30], and HoloLLM for 'Video' and 'mmWave' modalities. (b) Visualization of multimodal tokens from 2 action categories (denoted by different colors) for 'mmWave' (circles), 'WiFi' (pentagrams), and 'Text' (triangles) modalities generated by HoloLLM without or with UMIP.

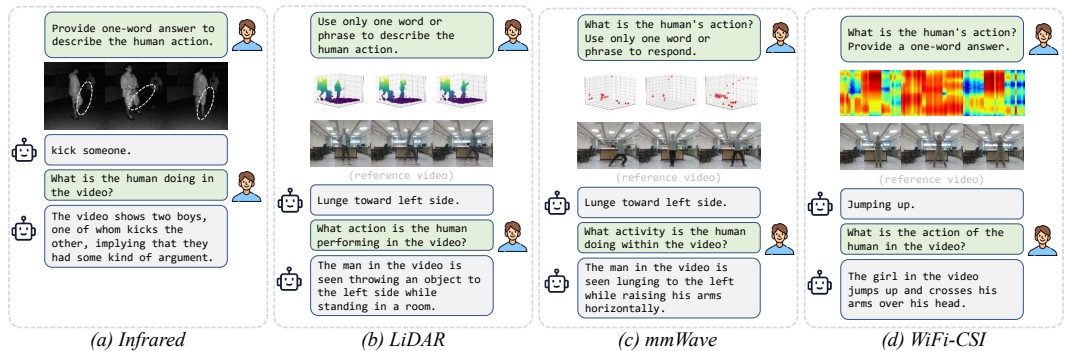

Figure 7: Qualitative results on sensing modalities. All examples are from the testing set of MM-Fi and XRF55 datasets under the "Cross-Environment" setting.

## 4.4 Qualitative Results

To intuitively show the effectiveness of HoloLLM, we visualize the aligned multimodal and text tokens of different models in Fig. 6 and show the qualitative results for sensing modalities in Fig. 7.

**Visualization Analysis of Multimodal and Text Tokens.** We randomly select 5 action categories from the MM-Fi dataset and visualize the aligned multimodal tokens generated by 'Baseline', OneLLM [30], and HoloLLM. As shown in Fig. 6 (a), only the multimodal tokens generated by HoloLLM are well grouped based on action categories across all modalities. This shows that tailored encoders are essential for capturing modality-specific discriminative features, while HoloLLM effectively preserves them via UMIP. Besides, we present the tokens of two sensing modalities (mmWave and WiFi) and the tokens of text captions for two action categories in Fig. 6 (b). Compared with only adopting the tailored encoders and Q-former ('HoloLLM w/o UMIP'), the multimodal tokens generated by HoloLLM can better align with the ground-truth text captions for different action categories. It intuitively shows that our UMIP helps achieve better multimodal alignment with text.

**Qualitative Results.** We give some qualitative results of HoloLLM in the "CrossEnv" setting in Fig. 7. These results show that HoloLLM can perform Action QA and Action Caption tasks across sensing modalities in diverse environments. More results and analysis are detailed in Appendix C.

## 5    Conclusion and Limitations

In this work, we present HoloLLM, an MLLM that integrates rare but powerful sensing modalities to enable seamless human perception and reasoning across heterogeneous real-world scenarios. Based on limited data, we propose the Universal Modality Injection Projector (UMIP) to efficiently align sensing modalities with the text via only minimal fine-tuning. Besides, the modality-specific discriminative features are adequately explored by tailored encoders and adaptively injected into the aligned multimodal tokens through UMIP. Thanks to UMIP, HoloLLM shows significant improvements compared to other state-of-the-art MLLMs on our multisensory language-grounded benchmarks.

**Limitation and Future work.** As the first attempt to human perception and reasoning based on sensing modalities, our work is limited to human action recognition, question answering, and caption. However, MLLMs needed to support more tasks to meet the requirements of real-world applications, such as task planning and agent action generation, which will be explored in our future work.

## Acknowledgments and Disclosure of Funding

This work is supported by a Start-up Grant from Nanyang Technological University and jointly funded by the Singapore Ministry of Education (MOE) under a Tier-1 research grant.

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

# Appendix

## A    Technical Details on Data Curation

In this section, we elaborate on more details about the data generation process for action question answering (QA) and captions.

### A.1    Action QA generation

We formulate Action QA as a question answering task with options. To ensure precision, two seed questions are first annotated by a human expert, as shown in Fig. 8. We then adopt GPT-4o [35] to

> **Human expert annotated seed questions**
>
> - "What is the human's action according to the input data? Answer the question by selecting a word or phrase from the attached Action List."
>
> - "Please select a word or phrase from the Action List which could represent the action of the human in the input data."

Figure 8: Two seed questions for action QA annotated by human experts.

rewrite the seed questions to enhance the diversity, which extends them to a list of 15 questions. Here, the utilized prompt and 5 examples of rewritten questions are listed in Fig. 9.

Afterwards, for each data sample from a certain modality, a question is randomly selected from the list to construct the QA question ('*Question*'). Furthermore, all action categories from the sensing dataset, along with the action label of the data sample, are appended to the question to generate the *Action List* (options) and the corresponding *Answer*. As shown in Fig. 10, a typical Action QA data sample can be formulated as: {*Question*, *Action List*, *Answer*} along with the input videos from diverse modalities (The vision modality is shown in Fig. 10). During training and testing, the input videos combined with *Question* and *Action List* are passed to the HoloLLM to predict the *Answer*.

### A.2    Action Caption generation

The textual descriptions of human actions are not available in both the MM-Fi and XRF55 datasets. Therefore, the captions cannot be directly generated using in-context learning. To this end, we propose a human-VLM collaborative pipeline to address caption generation. Specifically, we evenly select 108 / 110 samples from the MM-Fi / XRF55 dataset across various environments, subjects, and action categories to form seed data samples. These seed samples are captioned by human experts,

**Prompting GPT-4 to rewrite questions**

**Prompts:**

"You are a professional writer. Now I want you to do sentence rewriting task. I will give you a sentence enclosed in parentheses, you give me ten rewriting sentences. You cannot change the original meaning of the sentence. On this basis, you should try to make the rewritten sentence structure as diverse as possible."

- - - - - - - - - - - - - - - - - - - - - - - - - - - - - - - - - - - - - - - - - - - - - - - - - - - - - - -

**Rewritten Questions:**

- "Based on the input data, what action is the human taking? Select a word or phrase from the attached Action List."

- "What action does the input data indicate the human is performing? Choose a term from the Action List to answer."

- "Identify the human's action from the input data by selecting a word or phrase from the Action List provided."

- "According to the input data, what is the human action? Select an appropriate term from the Action List."

- "Find a word or phrase in the Action List that can indicate the human's action according to the input data."

Figure 9: Prompting GPT-4o [35] to rewrite the seed questions to enhance diversity. We provide 5 questions rewritten by GPT-4o as examples.

**Data sample of Action QA**

**Question:**

"What is the human's action according to the input data? Answer the question by selecting a word or phrase from the attached Action List."

**Action List:**

"Action List: 'stretching and relaxing', 'horizontal chest expansion', 'vertical chest expansion', 'left twist', 'right twist', 'mark time', 'left limb extension', 'right limb extension', 'lunge toward left-front', 'lunge toward right-front', 'both limb extension', 'squat', 'raising left hand', 'raising right hand', 'lunge toward left side', 'lunge toward right side', 'waving left hand', 'waving right hand', 'picking up things', 'throwing toward left side', 'throwing toward right side', 'kicking toward left side', 'kicking toward right side', 'left body extension', 'right body extension', 'jumping up', 'bowing'"]"

**Input Video**

**Action Label:** 'bowing'.

Figure 10: A typical data sample for the Action QA task.

and each caption is rewritten by GPT-4o as detailed in Appendix A.1 to enhance diversity, resulting in 5 captions for each seed sample. We then transfer a seed sample to an in-context caption sample: {*Question*, *Video*, *Caption*} by (1) Add a fixed question: *Please give detailed descriptions of human's action.*, (2) Attach the video sequence of the seed sample, and (3) Randomly select a caption. A typical in-context caption sample is shown in Fig. 11 (a).

Consequently, for the remaining data samples in both datasets, we adopt LLaVA-Video [26] to automatically generate captions through in-context learning. As illustrated in Fig. 12, for a specific data sample $s$, we first build the prompting `message` by progressively integrating the 'system prompt' and an in-context caption sample of the same category as $s$. The in-context caption sample contains input `sample['video']` with `sample['question']` and output `sample['response']`. Then, the `query['video']` and `query['question']` are appended to the `message` as the sample to be captioned. The complete prompting `message` is passed to the LLaVA-Video to obtain the *Caption*. As shown in Fig. 11 (b), a data sample for the Action Caption task can be formulated as: {*Question*, *Caption*}, which is shared across all modalities. During training and testing, the input videos combined with *Question* are passed to the HoloLLM to generate the *Caption*.

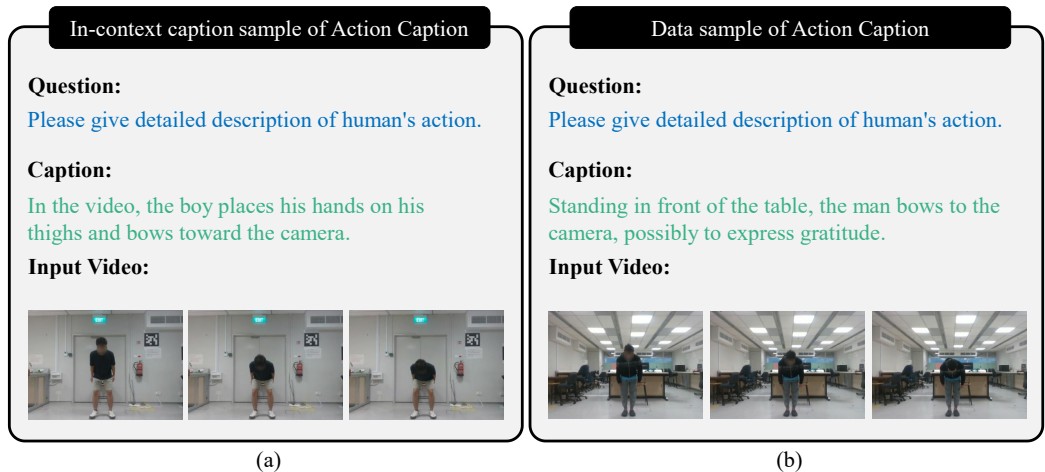

Figure 11: (a) A human-annotated in-context caption sample. (b) A typical data sample for the Action Caption task.

```
messages = [ {"role":"system", "content": """You are an AI visual assistant, and you are seeing a
video. You are going to do human action caption task based on the video. You will get a video and the Action
Label for the human action in the video. The Action Label will be enclosed in a pair of brackets. You need to give
helpful, detailed, and accurate descriptions on the human action.

Here are some examples. You need to generate a new description based on your observation on the given
examples."""}]
for sample in in_context_caption_samples:
        messages.append({"role":"user", "content": [sample['video'], sample['question']]})
        messages.append({"role":"assistant", "content": sample['responses']})

messages.append({"role":"user", "content": [query['video'], query['question']]})
```

Figure 12: Prompting `messages` passed to LLaVA-Video [26] to automatically generate caption for sensing data. Human-annotated in-context caption samples are included in the prompt, where each sample has input `sample['video']` with `sample['questions']` and output `sample['response']`.

## B  Details on Experimental Settings

In this section, we provide more details on the experimental settings. Specifically, we first present more details for the datasets in Appendix B.1. Followed by detailed statistics for three experimental settings in Appendix B.2. Finally, the training details of HoloLLM are elaborated in Appendix B.3.

### B.1  Datasets

**MM-Fi.**  MM-Fi consists of 27 action categories and 40 human subjects from 4 different environments. Each human subject contains synchronized sequences with aspects of 5 modalities: Video (V), Depth images (D), LiDAR point clouds (L), mmWave point clouds (M), and WiFi-CSI (W). In total, there are 16,448 multimodal sequences in the MM-Fi datasets, and 5 frames are evenly sampled from each sequence to form a data sample.

**XRF55.**  For XRF55, we only consider the human subjects with video modality, resulting in 19 human subjects from 4 different environments with 55 action categories. Each human subject contains synchronized sequences with aspects of 5 modalities: Video (V), Depth images (D), Infrared images

(I), RFID signals (R), and WiFi-CSI (W). In total, there are 19,800 multimodal sequences in the XRF55 datasets, and 10 frames are evenly sampled from each sequence to form a data sample.

## B.2 Benchmark

We design three experimental settings: (1) Random Split (Random), (2) Cross-Subject Split (CrossSub), and (3) Cross-Environment Split (CrossEnv). Specifically, 'Random' involves a random split of all samples with a ratio of 3:1, and 'CrossSub' / 'CrossEnv' selects samples from nonoverlapping human subjects / environments for training and testing.

| Settings | MM-Fi | | XRF55 | |
|---|---|---|---|---|
| | Train Size | Test Size | Train Size | Test Size |
| Random | 12,336 | 4,112 | 14,850 | 4,950 |
| CrossSub | 11,657 | 4,791 | 14,300 | 5,500 |
| CrossEnv | 12,565 | 3,883 | 16,500 | 3,300 |

Table 4: The detailed statistics of three experimental settings of HoloLLM.

Detailed statistics for three experimental settings on the sizes of the training and testing sets are summarized in Tab. 4.

## B.3 Training Details

The AdamW optimizer with $\beta_1 = 0.9$, $\beta_2 = 0.95$, and weight decay of 0.1 is adopted in our training. For stage one, we utilize the training set of the corresponding experimental setting (Random, CrossSub, CrossEnv) to pretrain the tailored encoders for 120 epochs on a single A100 GPU. The learning rate is initialized to 0.1 with a linear warmup strategy for 10 epochs, and then decayed at the 60-th and 100-th epochs with a decay factor of 0.1. For stage two, we train the HoloLLM on 2 A100 GPUs for 5 epochs. We set the accumulated iterations to 4 and form an effective batch size of 64 / 48 for the MM-Fi / XRF55 datasets, respectively. Followed by OneLLM [30], the linear warmup strategy is utilized for the first 2K iterations with a maximum learning rate of 2e-5.

| Settings | Models | Sources | Types | MM-Fi | | | | | | XRF55 | | | | | |
|---|---|---|---|---|---|---|---|---|---|---|---|---|---|---|---|
| | | | | V | D | M | L | W | Avg | V | D | I | R | W | Avg |
| Random | Tokenpacker | arXiv'24 | Proj | 15.6 | 14.5 | 32.8 | 18.9 | 11.5 | 18.7 | 9.2 | 8.2 | 7.1 | 2.2 | 6.5 | 6.6 |
| | Honeybee | CVPR'24 | Proj | 9.0 | 11.3 | 27.8 | 12.6 | 10.3 | 14.2 | 2.9 | 4.3 | 3.8 | 1.9 | 7.4 | 4.0 |
| | OneLLM | CVPR'24 | Proj | 9.0 | 9.0 | 16.2 | 6.1 | 9.0 | 9.9 | 2.1 | 2.0 | 1.5 | 1.9 | 2.3 | 2.0 |
| | ImageBind | CVPR'23 | Enc | 99.5 | 96.8 | 63.7 | 13.2 | 11.7 | 57.0 | 90.8 | 59.1 | 93.3 | 14.6 | **51.1** | 61.8 |
| | HoloLLM | - | Proj | **99.8** | **99.7** | **97.3** | **92.8** | **83.5** | **86.5** | **98.0** | **96.9** | **97.5** | **39.1** | 37.7 | **73.8** |
| CrossSub | Tokenpacker | arXiv'24 | Proj | 24.1 | 14.9 | 33.9 | 12.9 | **11.1** | 19.4 | 8.6 | 8.9 | 6.3 | 2.8 | 4.8 | 6.3 |
| | Honeybee | CVPR'24 | Proj | 11.6 | 9.3 | 27.2 | 13.9 | 8.3 | 14.1 | 3.0 | 4.2 | 4.3 | 2.4 | 4.9 | 3.8 |
| | OneLLM | CVPR'24 | Proj | 8.3 | 8.3 | 18.9 | 6.6 | 8.3 | 10.1 | 2.0 | 1.8 | 1.8 | 2.2 | 2.6 | 2.1 |
| | ImageBind | CVPR'23 | Enc | 85.7 | 70.7 | 56.9 | 17.7 | 11.0 | 48.4 | 18.6 | 14.4 | 22.0 | 4.2 | **5.7** | 13.0 |
| | HoloLLM | - | Proj | **98.4** | **99.0** | **93.8** | **76.6** | 9.3 | **75.4** | **46.4** | **47.8** | **42.2** | **4.4** | 4.5 | **29.1** |
| CrossEnv | Tokenpacker | arXiv'24 | Proj | 15.0 | 9.4 | 20.2 | 11.7 | 10.7 | 13.4 | 9.8 | 10.1 | 6.2 | 2.0 | 4.5 | 6.5 |
| | Honeybee | CVPR'24 | Proj | 13.5 | 12.9 | 26.7 | 12.8 | **11.2** | 15.4 | 8.9 | 4.9 | 3.1 | 1.4 | **6.7** | 5.0 |
| | OneLLM | CVPR'24 | Proj | 9.5 | 9.5 | 5.6 | 6.8 | 2.4 | 6.8 | 1.5 | 3.5 | 1.8 | 1.9 | 2.3 | 2.2 |
| | ImageBind | CVPR'23 | Enc | 45.7 | 8.0 | 26.8 | 8.6 | 9.5 | 19.7 | 6.7 | 6.8 | 21.1 | 2.5 | **2.5** | 7.9 |
| | HoloLLM | - | Proj | **80.6** | **93.2** | **61.0** | **53.0** | 9.5 | **59.5** | **28.9** | **12.4** | **28.3** | 1.7 | 2.5 | **14.8** |

Table 5: Evaluation of Human Action Recognition on MM-Fi [11] and XRF55 [10] across three settings. The Accuracy (%) is reported.

## C Additional Experiments

In this section, we present the quantitative results for Action Recognition in Appendix C.1, along with complete qualitative results and a comprehensive analysis for all modalities in Appendix C.2. Additionally, the results of a naive multimodal fusion strategy are provided in Appendix C.3.

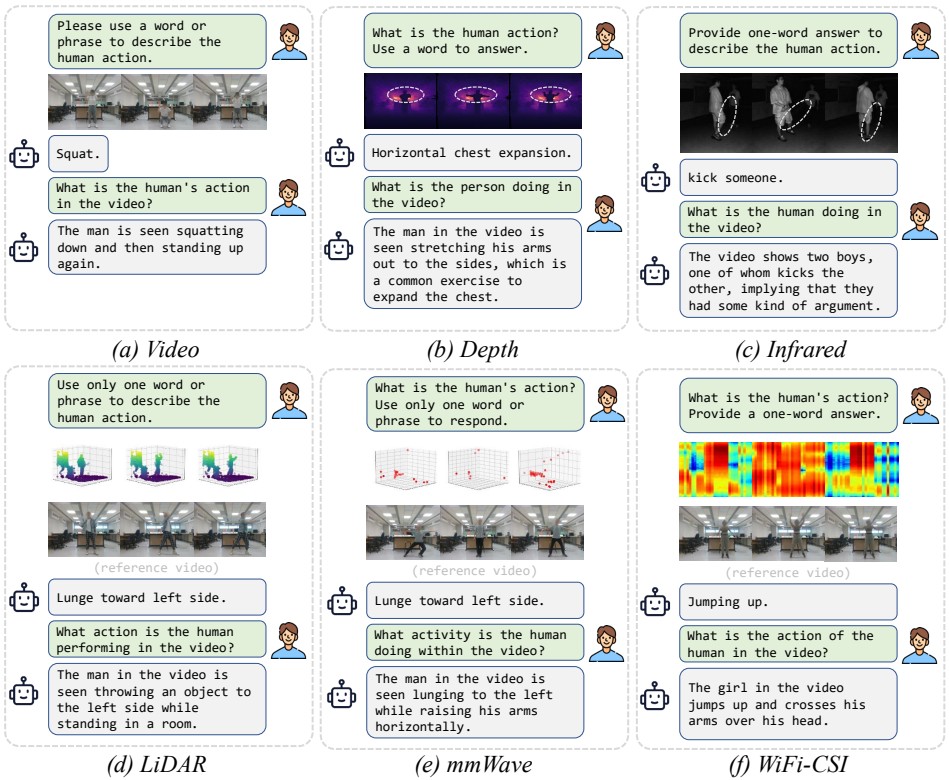

*(a) Video*    *(b) Depth*    *(c) Infrared*

*(d) LiDAR*    *(e) mmWave*    *(f) WiFi-CSI*

Figure 13: Qualitative results on sensing modalities. All examples are from the testing set of MM-Fi and XRF55 datasets under the "Cross-Environment" setting.

## C.1 Quantitative Results for Action Recognition

We provide the quantitative results corresponding to Fig. 5 for Action Recognition in Tab. 5. As discussed in Sec 4.2, HoloLLM is superior to other MLLMs on action recognition tasks for almost all modalities. For certain sensing modalities, comparable performances are observed under the "CrossSub" and "CrossEnv" settings. In fact, some sensing modalities are sensitive to different subjects and scenarios. Enhancing the generalization capability of sensing modalities toward diverse subjects and environments raises an important topic for the future.

## C.2 Complete Qualitative Results and Analysis

We show the complete qualitative results of HoloLLM under "CrossEnv" setting for both common and sensing modalities in Fig. 13. These results show that HoloLLM can perform Action QA and Action Caption tasks across multiple modalities in diverse environments. Moreover, the white dashed circles in Fig. 13 (b) and (c) illustrate that HoloLLM possesses the ability to capture fine-grained modality-specific action information and reason over it to identify the correct action category.

## C.3 Multimodal Fusion for HoloLLM

We conduct a preliminary exploration of HoloLLM's multimodal fusion capability. Specifically, we first adopt a naïve fusion strategy, in which aligned multimodal tokens generated by UMIP are directly concatenated. Multimodal reasoning is achieved by prepending the fused tokens to human instructions before feeding them into the LLM. For the MM-Fi and XRF55 datasets, we consider the Vision (V), mmWave (M), WiFi (W) and Vision (V), Infrared (I), WiFi (W) modalities, respectively. The results are summarized in Tab. 6.

| Modality | MM-Fi | | | Modality | XRF55 | | |
|---|---|---|---|---|---|---|---|
| | Recognition | QA | Caption | | Recognition | QA | Caption |
| V | 80.6 | 79.5 | 25.7 | V | 28.9 | 25.9 | 19.8 |
| M | 61.0 | 61.4 | 24.5 | I | 28.3 | 22.1 | 17.1 |
| W | 9.5 | 8.2 | 15.9 | W | 2.5 | 4.5 | 13.7 |
| V+M | **84.6** | 66.4 | 25.0 | V+I | **34.3** | 21.5 | 19.2 |
| V+W | 80.9 | 52.1 | 23.7 | V+W | 28.3 | 13.9 | 18.7 |
| M+W | 61.0 | 57.6 | 22.9 | I+W | 27.2 | 9.2 | 16.1 |
| V+M+W | **86.1** | 69.1 | 24.4 | V+I+W | **29.7** | 19.2 | 18.9 |

Table 6: Results of naïve multimodal fusion on MM-Fi and XRF55 under the "Cross-Environment" setting.

It shows naive multimodal fusion can enhance action recognition performance for certain modalities (highlighted in bold). However, for action QA and captioning tasks, naive multimodal fusion fails to improve performance.

Furthermore, we conduct experiments to explore more advanced adaptive fusion techniques, including weighted sum (WS) and max pooling (MP) across multimodal tokens. In the weighted-sum technique, the weights assigned to Vision and mmWave/Infrared modalities are 0.8 and 0.2, respectively.

| Modality | MM-Fi | | | Modality | XRF55 | | |
|---|---|---|---|---|---|---|---|
| | HAR | QA | Caption | | HAR | QA | Caption |
| V | 80.6 | 79.5 | 25.7 | V | 28.9 | 25.9 | 19.8 |
| M | 61.0 | 61.4 | 24.5 | I | 28.3 | 22.1 | 17.1 |
| V+M (Naïve) | **84.6** | 66.4 | 25.0 | V+I (Naïve) | **34.3** | 21.5 | 19.2 |
| V+M (WS) | 82.6 | **81.4** | **26.9** | V+I (WS) | 30.9 | **26.2** | **20.6** |
| V+M (MP) | 80.6 | 80.4 | 26.7 | V+I (MP) | 28.9 | 20.5 | 18.9 |

Table 7: Comparison of different multimodal fusion strategies on MM-Fi and XRF55 datasets.

As shown in Tab. 7, the weighted sum fusion strategy outperforms single-modality and naïve multimodal fusion on Action QA and captioning tasks. We believe that more advanced, learning-based multimodal fusion techniques could further improve performance. We expect HoloLLM to serve as a challenging benchmark that motivates the research community to develop deeper insights into multimodal fusion methods.

## C.4  Generalization for HoloLLM

HoloLLM generalizes to new modalities more efficiently than other MLLMs due to the novel UMIP architecture. Specifically, each new modality is equipped with a lightweight, modality-specific encoder that is extensively pretrained to capture fine-grained features. The new modality is then efficiently integrated into HoloLLM by UMIP with minor data and fine-tuning.

We demonstrate that HoloLLM achieves superior data efficiency over the 'Baseline' model on two novel modalities: Audio [41] and UWB [42]. The results are summarized in Tab. 8.

| | HAR | | Action QA | |
|---|---|---|---|---|
| | Audio | UWB | Audio | UWB |
| Baseline | 37.71 | 9.46 | 30.73 | 11.33 |
| HoloLLM | 63.94 | 16.77 | 54.17 | 16.45 |

Table 8: Results of HAR and Action QA tasks on novel Audio and UWB modalities.

Compared to the 'Baseline' model, HoloLLM can be efficiently generalized to novel modalities with minor data and fine-tuning.

## D   Broader Impacts

This work introduces HoloLLM, an MLLM that achieves seamless human perception and reasoning by integrating sensing modalities. HoloLLM establishes a multisensory foundation model, which is beneficial for developing embodied agents that are applicable in diverse real-world scenarios, including low-light environments, occlusions, and privacy-sensitive scenarios. However, HoloLLM may suffer similar concerns with other MLLMs, such as hallucinating or meaningless outputs (especially for some sensing modalities under cross-subject or cross-environment settings), inherited biases from base models, and energy consumption due to large-scale parameters. This raises important research topics such as enhancing the generalization capability of sensing modalities, aligning the base model with human intention, and efficiently pruning the foundation model. Despite these challenges, the release of HoloLLM would be beneficial, as it would foster further development of embodied AI.

