# OpenReview forum: "HoloLLM: Multisensory Foundation Model for Language-Grounded Human Sensing and Reasoning"
_NeurIPS.cc/2025/Conference — NeurIPS 2025 poster_

### Official Review · Reviewer_4VBc · 2025-06-09

**Clarity:** 3
**Significance:** 2
**Originality:** 2
**Rating:** 3
**Confidence:** 4

**Summary:**

This paper presents HoloLLM, a MLLM that takes multiple modalities for human activity recognition (and QA, captioning). The key technical components of this paper are: (1) modality-tailored (implemented as resnet of pointnet) encoder, and (2) UMIP, which uses CLIP vision encoder to iteratively perform cross-attention to refine the embedding. The authors also curated two multi-modality datasets, MM-Fi and XRF55, for sensory modality-text alignment, by generating captions or QAs according to the vision modality contained in the dataset.

**Questions:**

Please see weaknesses above.

**Ethical Concerns:**

["NO or VERY MINOR ethics concerns only"]

**Limitations:**

Please see weaknesses above.

**Paper Formatting Concerns:**

No.

**Quality:**

3

**Strengths And Weaknesses:**

Strengths:
1. HoloLLM achieved significantly better performance on the two datasets the authors curated.
2. The motivation is clearly stated. And the paper is well organized.
3. The authors contributed two curated datasets for sensory modality-text alignment experiments.

Weaknesses:
1. The technical contribution seems incremental, exhibiting limited originality or theoretical soundness. The ablation study showed that UMIP plays a less important role in the system, while most of the performance improvements are achieved by the tailored encoder, which is a resnet or pointnet encoder trained for each modality.
2. The contribution problem is amplified by the fact that the paper primarily focused on aligning single modality to text, with only preliminary multi-modal fusion results presented in appendix. As explained by the authors, since they only considered naive modality fusion methods, the multi-modal fusion results are only marginally better than single-model results. The authors may need to consider may more attention to multi-modal cases, given their motivation of "enable seamless human perception and reasoning across heterogeneous environments"

---

> ### Author Rebuttal · Authors · 2025-07-31
>
> We sincerely appreciate the reviewer 4VBc for insightful questions and suggestions. We answer all the questions by extensive empirical results and elaborations.
>
> **Q1: The technical contribution seems incremental, exhibiting limited originality or theoretical soundness.**
>
> **Answer:** We understand the reviewer may evaluate our work from the perspective of VLMs research community. However, we argue that HoloLLM bridges an underexplored gap in MLLM: how to introduce new rare modalities to enhance MLLM's capability for physical agents. HoloLLM’s target task, model architecture, training strategy, and benchmark results differ substantially from existing SOTA MLLMs. To this end, we provide a detailed comparison in the following table. As discussed in section 4.2, the Encoder-based methods (ImageBind [12], LanguageBind [16]) aim to unify multimodal embeddings, while the Projector-based methods (BLIP-2 [11], OneLLM [7]) focus on designing effective projectors.
>
> | |Task|Model Architecture|Training Strategy|Benchmark Results|
> |-|-|-|-|-|
> |HoloLLM (ours)|Align **data-scarce** but powerful **sensing modalities** (e.g., WiFi) with LLM using only limited data, while preserving their **heterogeneous features** for seamless human perception and reasoning.|A unified multimodal projector that considers **modality-specific features** by taking features from CLIP and **tailored encoders** as coarse queries and keys/values.|**Two-stage training.** (1) Pretrain tailored encoders to capture modality-specific features. (2) Fine-tune the UMIP to generate discriminative multimodal tokens aligned with text.|HoloLLM achieves SOTA performances on two **human sensing** datasets, establishing the first **multisensory benchmark** for human perception and reasoning.|
> |BLIP-2 [11]|**Image-text** alignment for visual-language tasks (e.g., image caption).|Original Q-former with **learnable queries** to integrate features from the vision encoder.|Pretraining projectors via **large-scale image-text** pairs.|Only evaluated on **image-language** benchmarks, such as VQAv2|
> |OneLLM [7]|Align **common modalities** (e.g., image, audio) with text for multimodal caption and QA tasks.|Q-former-style projector with multimodal **learnable queries** to integrate features from a unified encoder.| Pretraining projectors and further fine-tuning the whole LLM via **large-scale modality-text** pairs.|Only evaluated on **common modalities** benchmarks, such as VQAv2 (image), NextQA (video).|
> |ImageBind [12]|Unify embeddings of **common modalities** (e.g., text, audio) by aligning them with images.|Transformer encoders for common modalities.|Fine-tune all transformer encoders via contrastive learning on **large-scale modality-image** pairs.|Only evaluated on **common modalities non-reasoning** benchmarks (e.g., classification, retrieval), such as MSR-VTT (video), NYU-D (depth).|
> |LanguageBind [16]|Unify embeddings of **common modalities** (e.g., image, audio) by aligning them with text.|Transformer encoders with LoRA fine-tuning for common modalities.|Fine-tune all transformer encoders via contrastive learning and LoRA on **large-scale modality-image** pairs.|Only evaluated on **common modalities non-reasoning** benchmarks (e.g., classification, retrieval), such as MSR-VTT (video), AS-A (audio).|
>
> In summary, our proposed HoloLLM (1) deals with a more challenging and unexplored problem on data-scarce yet useful sensing modalities with heterogeneous features (**also mentioned by reviewers DvFs, 78xg, and mtFH**), (2) proposes a novel projector that incorporates modality-specific features for enhanced multimodal alignment (**also mentioned by reviewers DvFs and 78xg**), (3) presents a two-stage training strategy to sufficiently capture the modality-specific features and efficiently align multimodal inputs with the text space (**also mentioned by reviewer DvFs**), and (4) establishes the first multisensory benchmark for human perception and reasoning (**mentioned by all four reviewers**).
>
> **Q2: The ablation study showed that UMIP plays a less important role in the system, while most of the performance improvements are achieved by the tailored encoder.**
>
> **Answer:** We argue that UMIP is still critical given the two reasons:
>
> (1)   Significant improvement on the Action QA task (up to **9.8%** on MM-Fi).
>
> UMIP contributes to better aligning multimodal tokens with the text while sufficiently preserving discriminative, modality-specific features. This results in a substantial performance gain on Action QA tasks—up to 9.8% on MM-Fi, as shown in Tab. 3— which demand a deeper understanding of language-based human instructions and action categories. Additionally, Fig. 6 (b) provides an intuitive example of enhanced multimodal-text alignment, where multimodal tokens generated by UMIP and text tokens from ground-truth captions are correctly grouped by action category.
>
> (2) Additional experiments show that UMIP enhances the generalization capability of HoloLLM.
>
> UMIP adopts a **novel decoupled architecture**, where CLIP generates coarse queries and tailored encoders extract modality-specific features as keys and values. It enables HoloLLM to achieve improved zero-shot generalization and efficient adaptation to new modalities, as demonstrated by two additional experiments.
>
> (2a) Generalized to New Datasets.
>
> We report zero-shot results on SenseFi [8]. Specifically, SenseFi performs human sensing using WiFi-CSI signals and includes six novel action categories. We then apply the ‘Baseline’ model (as detailed in line 248) and HoloLLM trained on the MM-Fi dataset under the ‘random’ setting to perform zero-shot human action QA on the official SenseFi test set, which contains 264 samples.
>
> |Model|Action QA|
> |-|-|
> |Baseline|5.43|
> |HoloLLM|16.67|
>
> In zero-shot settings, HoloLLM can achieve better performance. The decoupled architecture of UMIP effectively leverages CLIP’s generalization capability—acquired through extensive pretraining—to generate coarse queries that generalize better than learnable queries.
>
> (2b) Generalization to Unseen Sensing Modalities
>
> HoloLLM can integrate new modalities more efficiently than other MLLMs, owing to the decoupled architecture of UMIP. Specifically, we extensively pretrain a lightweight, modality-specific, tailored encoder for each new modality to sufficiently capture fine-grained features. The new modality is then efficiently integrated into HoloLLM by UMIP with minor data and fine-tuning.
>
> We add an experiment to demonstrate the superior data efficiency of HoloLLM compared to the ‘Baseline’ model (as detailed in line 248). Specifically, we evaluate two novel modalities, audio [9] and UWB [10], with the results summarized below.
>
> | |HAR| |Action QA| |
> |-|:-:|:-:|:-:|:-:|
> | |Audio|UWB|Audio|UWB|
> |Baseline|37.71|9.46|30.73|11.33|
> |HoloLLM|63.94|16.77|54.17|16.45|
>
> Compared to the ‘Baseline’ model, HoloLLM can be efficiently generalized to novel modalities with minor data and fine-tuning.
>
> **Q3: The contribution problem is amplified by the fact that the paper primarily focused on aligning single modality to text. The authors may need to consider may more attention to multi-modal cases.**
>
> **Answer:** We appreciate the suggestion and believe this should be further illustrated. We deduce that the marginal improvement of naïve multimodal fusion arises from its failure to fully exploit complementary cross-modal information and reduce redundancy. As suggested, we consider more multi-modal fusion cases other than naive fusion (simply concatenate all multimodal tokens), including weighted sum (WS) and max pooling (MP) across multimodal tokens. The results of additional experiments are listed as follows.
>
> ||MM-Fi||||XRF55|||
> |-|-|-|-|-|-|-|-|
> | |HAR|QA|Caption| |HAR|QA|Caption|
> |V|80.6|79.5|25.7|V|28.9|25.9|19.8|
> |M|61.0|61.4|24.5|I|28.3|22.1|17.1|
> |V+M (Naive)|**84.6**|66.4|25.0|V+I (Naive)|**34.3**|21.5|19.2|
> |V+M (WS)|82.6|**81.38**|**26.9**|V+I (WS)|30.91|**26.24**|**20.6**|
> |V+M (MP)|80.6|80.43|26.7|V+I (MP)|28.9|20.48|18.9|
>
> The results show that the weighted sum fusion techniques outperform single-modality and naïve multimodal fusion on Action QA and captioning tasks. We believe that more advanced, learning-based multimodal fusion techniques could further enhance performance. We hope that HoloLLM can serve as a challenging benchmark to inspire the research community toward more insightful contributions in this multimodal fusion technique.
>
> **References**
>
> [1] XRF V2: A Dataset for Action Summarization with Wi-Fi Signals, and IMUs in Phones, Watches, Earbuds, and Glasses. (ACM IMWUT 2025)
>
> [2] Mm-fi: Multi-modal non-intrusive 4d human dataset for versatile wireless sensing. (NeurIPS 2023)
>
> [3] Domain‑Adversarial Training of Neural Networks. (JMLR 2016)
>
> [4] Meta-Transformer: A Unified Framework for Multimodal Learning. (arXiv 2023)
>
> [5] Chain‑of‑Thought Prompting Elicits Reasoning in Large Language Models. (NeurIPS 2022)
>
> [6] Measuring Nominal Scale Agreement among Many Raters. (Psychological Bulletin 1971)
>
> [7] OneLLM: One Framework to Align All Modalities with Language (CVPR 2024)
>
> [8] SenseFi: A library and benchmark on deep-learning-empowered WiFi human sensing (Patterns 2023)
>
> [9] SAMoSA: Sensing Activities with Motion and Subsampled Audio (ACM IMWUT 2022)
>
> [10] SleepPoseNet: Multi-view learning for sleep postural transition recognition using UWB (IJBHI 2020)
>
> [11] BLIP-2: Bootstrapping Language-Image Pre-training with Frozen Image Encoders and Large Language Models (ICML 2023)
>
> [12] Imagebind: One embedding space to bind them all (CVPR 2023)
>
> [13] Omnifusion: 360 monocular depth estimation via geometry-aware fusion (CVPR 2022)
>
> [14] UNIFIED-IO: A Unified Model for Vision, Language, and Multi-modal Tasks (ICLR 2023)
>
> [15] XRF55: A Radio Frequency Dataset for Human Indoor Action Analysis. (ACM IMWUT 2024)
>
> [16] LanguageBind: Extending Video-Language Pretraining to N-modality by Language-based Semantic Alignment (ICLR 2024)

---

> > ### Comment · Reviewer_4VBc · 2025-08-06
> >
> > Thank you for the detailed rebuttal and the substantial additional experiments, which I found comprehensive.
> >
> > That said, I still find the technical novelty of the paper to be somewhat limited. The current approach appears to be more of a direct adaptation of existing VLM techniques to the newly curated sensing datasets, rather than introducing fundamentally new methodological insights. VLMs also have encoders "tailored" for visual modalities (which may also be pretrained), and various kind of (possibly multi-modal) projectors that bridges visual embeddings with textual embeddings.

---

> > > ### Author Response · Authors · 2025-08-07
> > > **Response (1/2)**
> > >
> > > Thank you for your response. We are happy that the comprehensive empirical results have been appreciated.
> > >
> > > Regarding the technical novelty, we fully understand the concerns raised by the reviewer, yet we believe the argument that “the technical novelty of the paper is somewhat limited” is not reasonable. We respond to each argument as below.
> > >
> > > > The current approach appears to be more of a direct adaptation of existing VLM techniques to the newly curated sensing datasets, rather than introducing fundamentally new methodological insights.
> > >
> > > The first novelty of our work is a novel task, which is fundamentally different from existing Vision-Language Models (VLMs), and it is shown that “direct adaptation of existing VLM techniques to the newly curated sensing datasets” will result in **very poor performances** (as shown in Tab. 1 and Tab. 2 in the original manuscript).
> > >
> > > Our task aims to align **rare yet powerful and underexplored sensing modalities** with LLM to enable seamless human perception. The task is extremely difficult due to two key challenges:
> > >
> > > 1. Data scarcity. Sensing data is typically collected in labs, yielding only a few thousand samples. As a result, we must align sensing modalities with text using only limited training data.
> > >
> > > 2. Heterogeneous features. To model the physical world, different sensors are designed to leverage distinct physical characteristics (e.g., wavelengths and frequencies) at multiple granularities, resulting in extreme heterogeneity that must be considered in architecture designs.
> > >
> > > To formulate this task, we curate new datasets and propose a comprehensive benchmark. Our work serves as **a challenging benchmark to inspire the research community toward more insightful contributions in this multimodal LLM**.
> > >
> > > The second novelty is the methodology design. Specifically, we solve the challenges by introducing modality-specific tailored encoders with fundamentally novel UMIP architecture to achieve efficient multimodal alignment with minor data and fine-tuning. Compared to existing VLM techniques, our method can improve language-grounded human sensing accuracy by **up to 30%**.
> > >
> > > Furthermore, we have already compared HoloLLM with other SOTA VLMs and MLLMs in detail, **demonstrating significant differences** regarding the objectives, model architecture, training strategy, and benchmark results.
> > >
> > > | |Task|Model Architecture|Training Strategy|Benchmark Results|
> > > |-|-|-|-|-|
> > > |HoloLLM (ours)|Align **data-scarce** but powerful **sensing modalities** (e.g., WiFi) with LLM using only limited data, while preserving their **heterogeneous features** for seamless human perception and reasoning.|A unified multimodal projector that considers **modality-specific features** by taking features from CLIP and **tailored encoders** as coarse queries and keys/values.|**Two-stage training.** (1) Pretrain tailored encoders to capture modality-specific features. (2) Fine-tune the UMIP to generate discriminative multimodal tokens aligned with text.|HoloLLM achieves SOTA performances on two **human sensing** datasets, establishing the first **multisensory benchmark** for human perception and reasoning.|
> > > |BLIP-2 [4]|**Image-text** alignment for visual-language tasks (e.g., image caption).|Original Q-former with **learnable queries** to integrate features from the vision encoder.|Pretraining projectors via **large-scale image-text** pairs.|Only evaluated on **image-language** benchmarks, such as VQAv2|
> > > |OneLLM [3]|Align **common modalities** (e.g., image, audio) with text for multimodal caption and QA tasks.|Q-former-style projector with multimodal **learnable queries** to integrate features from a unified encoder.| Pretraining projectors and further fine-tuning the whole LLM via **large-scale modality-text** pairs.|Only evaluated on **common modalities** benchmarks, such as VQAv2 (image), NextQA (video).|
> > > |ImageBind [13]|Unify embeddings of **common modalities** (e.g., text, audio) by aligning them with images.|Transformer encoders for common modalities.|Fine-tune all transformer encoders via contrastive learning on **large-scale modality-image** pairs.|Only evaluated on **common modalities non-reasoning** benchmarks (e.g., classification, retrieval), such as MSR-VTT (video), NYU-D (depth).|
> > > |LanguageBind [14]|Unify embeddings of **common modalities** (e.g., image, audio) by aligning them with text.|Transformer encoders with LoRA fine-tuning for common modalities.|Fine-tune all transformer encoders via contrastive learning and LoRA on **large-scale modality-image** pairs.|Only evaluated on **common modalities non-reasoning** benchmarks (e.g., classification, retrieval), such as MSR-VTT (video), AS-A (audio).|

---

> > > > ### Author Response · Authors · 2025-08-07
> > > > **Response (2/2)**
> > > >
> > > > > VLMs also have encoders "tailored" for visual modalities (which may also be pretrained), and various kind of (possibly multi-modal) projectors that bridges visual embeddings with textual embeddings.
> > > >
> > > > Our method, including the encoders, is tailored to address the challenges beneath the novel task - how to incorporate rare yet powerful modalities into an MLLM. It’s not fair to judge the technical novelty by simply comparing the model architecture.
> > > >
> > > > For example, many landmark works employ similar architecture to previous works, but solve novel and important problems in their fields:
> > > >
> > > > - ViT [8] simply adopts the transformer architecture [9] in vision tasks.
> > > >
> > > > - Diffusion Policy [10] applies the DDPM [11] to generate action trajectories for robotic tasks.
> > > >
> > > > - LLaVA [1] essentially addresses the vision-language tasks (e.g., image caption) by simply connecting the CLIP vision encoder [12] and LLM.
> > > >
> > > > - PointLLM [2] adopts a similar architecture to LLaVA for 3D reasoning.
> > > >
> > > > - OneLLM [3] aligns multiple common modalities to language using a Q-former projector like BLIP-2 [4].
> > > >
> > > > These works indeed adopt existing networks as part of their methods, but they **address significant problems and thus become the milestone papers** in image recognition, 3D vision, imitation learning, and VLM. **Similarly, HoloLLM addresses significant problems behind a novel task with simple yet effective designs.** The empirical results in Tab. 1 and Tab. 2 show the effectiveness (up to 30%) over existing SOTA VLMs and MLLMs. Furthermore, we do not find that existing VLMs already have heterogeneous “tailored” encoders to handle different modalities with different physical characteristics of signals, since VLM only has one vision modality apart from language.
> > > >
> > > > We sincerely request the reviewer to consider our work regarding the novel task, the comprehensive benchmark with curated datasets, and an effective method serving as the baseline.
> > > >
> > > >
> > > > **References**
> > > >
> > > > [1] Visual Instruction Tuning. (NeurIPS 2023)
> > > >
> > > > [2] PointLLM: Empowering Large Language Models to Understand Point Clouds. (ECCV 2024)
> > > >
> > > > [3] OneLLM: One Framework to Align All Modalities with Language. (CVPR 2024)
> > > >
> > > > [4] BLIP-2: Bootstrapping Language-Image Pre-training with Frozen Image Encoders and Large Language Models. (ICML 2023)
> > > >
> > > > [5] LLaMA-Adapter: Efficient Fine-tuning of Large Language Models with Zero-initialized Attention. (ICLR 2024)
> > > >
> > > > [6] mPLUG-Owl2: Revolutionizing Multi-modal Large Language Model with Modality Collaboration. (CVPR 2024)
> > > >
> > > > [7] ImageBind-LLM: Multi-modality Instruction Tuning. (arXiv 2023)
> > > >
> > > > [8] An Image is Worth 16x16 Words: Transformers for Image Recognition at Scale. (ICLR 2021)
> > > >
> > > > [9] Attention Is All You Need. (NeurIPS 2017).
> > > >
> > > > [10] Diffusion Policy: Visuomotor Policy Learning via Action Diffusion. (RSS 2023)
> > > >
> > > > [11] Denoising Diffusion Probabilistic Models. (NeurIPS 2020)
> > > >
> > > > [12] Learning Transferable Visual Models from Natural Language Supervision. (ICML 2021)
> > > >
> > > > [13] Imagebind: One embedding space to bind them all (CVPR 2023)
> > > >
> > > > [14] LanguageBind: Extending Video-Language Pretraining to N-modality by Language-based Semantic Alignment (ICLR 2024)

---

> > > > > ### Author Response · Authors · 2025-08-08
> > > > > **Looking forward to more discussions**
> > > > >
> > > > > Dear Reviewer 4VBc,
> > > > >
> > > > > As the deadline approaches, please feel free to let us know if you have any additional questions or concerns. Thank you for your time and consideration.
> > > > >
> > > > > Best Regards,
> > > > >
> > > > > Authors of Submission 19827

---

> ### Author Response · Authors · 2025-08-04
> **Looking forward to your reply**
>
> Dear Reviewer 4VBc,
>
> As the discussion phase deadline is approaching in **2 days**, could you please have a look at our response? We are looking forward to your reply!
>
> Feel free to let us know if you have any other concerns. Thanks for your time and effort!
>
> Best Regards,
>
> Authors of Submission 19827

---

> ### Author Response · Authors · 2025-08-05
> **Looking forward to your reply**
>
> Dear Reviewer 4VBc,
>
> As the discussion phase deadline is approaching in **1 day**, could you please have a look at our response? We are looking forward to your reply!
>
> Feel free to let us know if you have any other concerns. Thanks for your time and effort!
>
> Best Regards,
>
> Authors of Submission 19827

---

> > ### Comment · Area_Chair_qDXX · 2025-08-06
> > **Request to Respond to Author Rebuttal and Participate in Discussion**
> >
> > Dear Reviewer 4VBc,
> >
> > As the Area Chair for this paper, I would like to request your active participation in the ongoing discussion phase, especially since the paper has received mixed borderline reviews. Your contribution is essential to reaching a well-informed and balanced decision. Additionally, I would like to clarify the requirements related to the Mandatory Acknowledgment process. To fulfill this requirement, reviewers are expected to:
> >
> > (i) Carefully read the author rebuttal,
> >
> > (ii) Engage in meaningful discussion with the authors—and preferably also with fellow reviewers.
> >
> > (iii) Ask questions, consider responses, and actively participate in the exchange,
> >
> > (iv) Clearly articulate any unresolved concerns to give authors a fair opportunity to respond. Please avoid situations where the discussion implies “everything is great,” but the final justification form states otherwise. The discussion phase is designed to surface and clarify such issues.
> >
> > Kindly note that clicking the “Mandatory Acknowledgment” checkbox prematurely does not exempt reviewers from participating in the discussion. Reviewers who do not contribute meaningfully may be flagged using the “Insufficient Review” button, in line with this year’s responsible reviewing guidelines.
> >
> > Thank you for your time and thoughtful contributions to the review process.

---

### Official Review · Reviewer_mtFH · 2025-06-29

**Clarity:** 3
**Significance:** 3
**Originality:** 3
**Rating:** 4
**Confidence:** 3

**Summary:**

The paper introduces HoloLLM Multimodal Large Language Model designed to understand human behavior by integrating unconventional sensing modalities with natural language. This model addresses the limitations of Vision-Language Models (VLMs), which struggle with poor lighting, occlusions, or privacy concerns.

**Questions:**

Also in Weaknesses.
How is the proposed method similar / different from LanguageBind[1]? How would the model perform by directly applying Languagebind [1] technique?

[1] Zhu B, Lin B, Ning M, et al. Languagebind: Extending video-language pretraining to n-modality by language-based semantic alignment[J].

**Ethical Concerns:**

["NO or VERY MINOR ethics concerns only"]

**Final Justification:**

Most of my concerns are addressed. I will keep my positive score

**Limitations:**

Yes

**Paper Formatting Concerns:**

No concerns

**Quality:**

3

**Strengths And Weaknesses:**

Strength:
* This paper tackles a novel task of integrating and working with unconventional sensors
* Reasonable experiments demonstrate the effectiveness of the proposed method over existing works
* The paper is clearly written and easy to follow

Weakneses:
* How is the proposed method similar / different from LanguageBind[1]? How would the model perform by directly applying Languagebind [1] technique?
* The proposed method fails to improve performance at times.  For example, in the MM-Fi dataset, the single-modality vision model achieves 79.5% on Action QA, while the fused Vision+mmWave+WiFi model drops to 69.1% Which begs the usefulness of the proposed method and task.
* The proposed method shows poor generalization and cross-subject testing performance. As the paper claims that the model enables "seamless human perception and reasoning across heterogeneous environments". This begs the feasibility of its real-world utility.


[1] Zhu B, Lin B, Ning M, et al. Languagebind: Extending video-language pretraining to n-modality by language-based semantic alignment[J].

---

> ### Author Rebuttal · Authors · 2025-07-31
>
> We sincerely appreciate the reviewer mtFH for the detailed review and suggestions. Here we add extensive experiments and answer all questions.
>
> **Q1 How is the proposed method similar/different from LanguageBind[16]? How would the model perform by directly applying Languagebind [16] technique?**
>
> **Answer:** We appreciate the suggestion to compare HoloLLM with the recent unified multimodal encoder, LanguageBind [16], and apologize for having overlooked this work during our literature review. Here, we illustrate the differences regarding the target task, model architecture, training strategy, and benchmark results among ImageBind [12], LanguageBind, and HoloLLM as follows.
>
> | |Task|Model Architecture|Training Strategy|Benchmark Results|
> |-|-|-|-|-|
> |HoloLLM (ours)|Align **data-scarce** but powerful **sensing modalities** (e.g., WiFi) with LLM using only limited data, while preserving their **heterogeneous features** for seamless human perception and reasoning.|A unified multimodal projector that considers **modality-specific features** by taking features from CLIP and **tailored encoders** as coarse queries and keys/values.|**Two-stage training.** (1) Pretrain tailored encoders to capture modality-specific features. (2) Fine-tune the UMIP to generate discriminative multimodal tokens aligned with text.|HoloLLM achieves SOTA performances on two **human sensing** datasets, establishing the first **multisensory benchmark** for human perception and reasoning.|
> |ImageBind [12]|Unify embeddings of **common modalities** (e.g., text, audio) by aligning them with images.|Transformer encoders for common modalities.|Fine-tune all transformer encoders via contrastive learning on **large-scale modality-image** pairs.|Only evaluated on **common modalities non-reasoning** benchmarks (e.g., classification, retrieval), such as MSR-VTT (video), NYU-D (depth).|
> |LanguageBind [16]|Unify embeddings of **common modalities** (e.g., image, audio) by aligning them with text.|Transformer encoders with LoRA fine-tuning for common modalities.|Fine-tune all transformer encoders via contrastive learning and LoRA on **large-scale modality-image** pairs.|Only evaluated on **common modalities non-reasoning**
>
> In summary, the differences between our proposed HoloLLM and the LanguageBind are (1) deals with a more challenging problem on data-scarce yet powerful sensing modalities with heterogeneous features, (2) proposes a novel projector that incorporates fine-grained, modality-specific features for enhancing multimodal alignment, (3) presents a two-stage training strategy to sufficiently capture the modality-specific features and efficiently align multimodal inputs with the text space, and (4) establishes the first multisensory benchmark for human perception and reasoning.
>
> As suggested, we evaluate LanguageBind. Similar to ImageBind, we treat it as a strong multimodal encoder, connect it to LLaMA2-7B, and fine-tune it on both MM-Fi and XRF55 same as HoloLLM. The results are summarized as follows (Top: MM-Fi, Bottom: XRF55).
>
> |Settings|Models|Sources|Types|Action QA| | | | | |Action Caption| | | | | |
> |-|-|-|-|-|-|-|-|-|-|-|-|-|-|-|-|
> | | | | |V|D|M|L|W|A|V|D|M|L|W|A|
> |Random|LanguageBind|ICLR’24|Enc|7.2|7.5|7.2|3.6|5.4|6.2|16.5|15.5|14.1|15.5|16.3|15.6|
> | |ImageBind|CVPR’23|Enc|89.3|76.7|45.8|11.1|8.0|46.2|28.4|21.2|18.8|16.2|14.8|19.9|
> | |HoloLLM|-|Proj|99.8|99.7|95.8|84.2|52.8|86.5|30.8|31.1|29.6|27.4|23.0|28.4|
> |Cross-Sub|LanguageBind|ICLR’24|Enc|6.8|6.1|6.5|3.3|4.9|5.5|14.9|15.2|11.1|15.1|16.1|14.5|
> | |ImageBind|CVPR’23|Enc|76.9|43.3|45.5|6.8|7.7|36.0|25.8|21.0|20.9|15.3|16.5|19.9|
> | |HoloLLM|-|Proj|98.0|98.9|88.0|66.5|8.0|71.9|30.6|30.5|29.5|24.9|16.7|26.4|
> |Cross-Env|LanguageBind|ICLR’24|Enc|6.7|5.9|4.1|6.3|5.2|5.6|16.2|15.4|5.7|5.7|12.3|11.1|
> | |ImageBind|CVPR’23|Enc|41.0|5.3|24.0|7.6|5.5|16.7|19.4|19.8|17.6|15.0|14.9|17.3|
> | |HoloLLM|-|Proj|79.5|91.6|61.4|41.4|8.2|56.4|25.7|27.5|24.5|19.6|15.9|22.6|
>
>
> |Settings|Models|Sources|Types|Action QA| | | | | |Action Caption| | | | | |
> |-|-|-|-|-|-|-|-|-|-|-|-|-|-|-|-|
> | | | | |V|D|I|R|W|A|V|D|I|R|W|A|
> |Random|LanguageBind|ICLR’24|Enc|27.7|10.9|38.1|2.5|5.7|17.0|13.8|15.2|12.3|13.6|14.7|13.9|
> | |ImageBind|CVPR’23|Enc|62.2|22.2|79.0|5.3|10.0|35.8|19.3|13.0|24.3|12.3|12.7|16.3|
> | |HoloLLM|-|Proj|94.5|92.3|92.6|27.1|11.2|63.5|34.2|34.8|34.7|15.5|14.0|26.6|
> |Cross-Sub|LanguageBind|ICLR’24|Enc|4.7|9.7|2.2|2.5|5.2|4.9|14.5|14.6|12.5|13.7|13.9|13.8|
> | |ImageBind|CVPR’23|Enc|13.3|11.1|20.4|3.8|4.9|10.7|15.8|14.6|18.3|12.7|14.5|15.2|
> | |HoloLLM|-|Proj|44.3|42.1|38.3|3.4|3.6|26.3|22.3|23.1|22.8|11.8|13.7|18.7|
> |Cross-Env|LanguageBind|ICLR’24|Enc|2.8|2.5|2.0|2.1|3.4|2.6|12.8|12.5|13.5|13.5|12.7|13.0|
> | |ImageBind|CVPR’23|Enc|4.7|4.9|16.9|2.8|2.6|6.4|13.1|14.3|16.9|12.7|12.0|13.8|
> | |HoloLLM|-|Proj|25.9|8.6|22.1|2.6|4.5|12.8|19.8|14.7|17.1|10.8|13.7|15.2|
>
> Compared to ImageBind, LanguageBind shows relatively lower performance. We deduce there are two main factors: (1) ImageBind employs a stronger vision encoder (CLIP ViT-H), and (2) it fully fine-tunes modality-specific encoders (e.g., depth), whereas LanguageBind only adopts CLIP ViT-L and applies LoRA for fine-tuning. We will incorporate the results into Tab 1 and Tab 2 in the final manuscript.
>
> **Q2: The proposed method fails to improve performance on Action QA with single/fusion setting. Illustrate the usefulness of the method and task.**
>
> **Answer:** We appreciate the comment regarding the multimodal fusion capabilities of HoloLLM. As mentioned, the results in Appendix C.3 show that naïve multimodal fusion (simply concatenating all multimodal tokens) fails to improve performance on action QA and captioning tasks. We conjecture that this is because naïve multimodal fusion does not fully leverage complementary information across modalities nor effectively reduce redundancy. Moreover, we add experiments to explore stronger adaptive fusion techniques, including weighted sum (WS) and max pooling (MP) across multimodal tokens.
>
> ||MM-Fi||||XRF55|||
> |-|-|-|-|-|-|-|-|
> | |HAR|QA|Caption| |HAR|QA|Caption|
> |V|80.6|79.5|25.7|V|28.9|25.9|19.8|
> |M|61.0|61.4|24.5|I|28.3|22.1|17.1|
> |V+M (Naive)|**84.6**|66.4|25.0|V+I (Naive)|**34.3**|21.5|19.2|
> |V+M (WS)|82.6|**81.38**|**26.9**|V+I (WS)|30.91|**26.24**|**20.6**|
> |V+M (MP)|80.6|80.43|26.7|V+I (MP)|28.9|20.48|18.9|
>
> The results show that the weighted sum fusion techniques outperform single-modality and naïve multimodal fusion on Action QA and captioning tasks. We believe that more advanced, learning-based multimodal fusion techniques could further enhance performance. We hope that HoloLLM can serve as a challenging benchmark to inspire the research community toward more insightful contributions in this multimodal fusion technique.
>
> **Q3: The proposed method shows poor generalization and cross-subject testing performance, which may hinder the real-world feasibility.**
>
> **Answer:** In fact, the cross-subject and cross-env generalization of these modalities (e.g., WiFi) is very challenging, as theoretically they are susceptible to variations in subjects and environments, and the data scarcity makes these tasks even harder. Though the tasks are hard, HoloLLM still outperforms other SOTA MLLMs in both settings by a large margin. We further conduct additional experiments to demonstrate the generalization capabilities of HoloLLM from two additional perspectives: new datasets and unseen sensing modalities:
>
> (1) Generalized to New Datasets.
>
> We report zero-shot results on SenseFi [8]. Specifically, SenseFi performs human sensing using WiFi-CSI signals and includes six novel action categories. We then apply the ‘Baseline’ model (as detailed in line 248) and HoloLLM trained on the MM-Fi dataset under the ‘random’ setting to perform zero-shot human action QA on the official SenseFi test set, which contains 264 samples.
>
> |Model|Action QA|
> |-|-|
> |Baseline|5.43|
> |HoloLLM|16.67|
>
> In zero-shot settings, HoloLLM’s superior performance demonstrates its strong generalization ability and discriminability.
>
> (2) Generalization to Unseen Sensing Modalities
>
> HoloLLM can integrate new modalities more efficiently than other MLLMs, owing to the novel architecture of UMIP. Specifically, we extensively pretrain a lightweight, modality-specific, tailored encoder for each new modality to sufficiently capture fine-grained features. The new modality is then efficiently integrated into HoloLLM by UMIP with minor data and fine-tuning.
>
> We add an experiment to demonstrate the superior data efficiency of HoloLLM compared to the ‘Baseline’ model (as detailed in line 248). Specifically, we evaluate two novel modalities, audio [9] and UWB [10], with the results summarized below.
>
> | |HAR| |Action QA| |
> |-|:-:|:-:|:-:|:-:|
> | |Audio|UWB|Audio|UWB|
> |Baseline|37.71|9.46|30.73|11.33|
> |HoloLLM|63.94|16.77|54.17|16.45|
>
> Compared to the ‘Baseline’ model, HoloLLM can be efficiently generalized to novel modalities with minor data and fine-tuning.
>
> In future work, we aim to enhance the generalization ability of HoloLLM by constructing a large-scale sensing dataset. Specifically,  we plan to integrate existing public datasets [1,2] and collect additional data from a broader range of environments (e.g., outdoor) and tasks (e.g., task planning and agent action generation). By fine-tuning HoloLLM with this new dataset, we believe that HoloLLMv2 will exhibit significantly improved cross-subject and cross-environment generalization. This work focuses on aligning new yet powerful sensing modalities with LLM, given limited data and various modalities, and thus offers a novel technical solution with a comprehensive benchmark. Regarding the real-world feasibility, we believe introducing complementary modalities is critical for embodied AI agents. HoloLLM denotes a baseline to engage the research community in further advancing MLLMs with data-scarce data modality.
>
> **References**
>
> Due to limited characters, please refer to the responses of Reviewer 4VBc for references.

---

> ### Author Response · Authors · 2025-08-04
> **Looking forward to your reply**
>
> Dear Reviewer mtFH,
>
> As the discussion phase deadline is approaching in **2 days**, could you please have a look at our response? We are looking forward to your reply!
>
> Feel free to let us know if you have any other concerns. Thanks for your time and effort!
>
> Best Regards,
>
> Authors of Submission 19827

---

> ### Author Response · Authors · 2025-08-05
> **Looking forward to your reply**
>
> Dear Reviewer mtFH,
>
> As the discussion phase deadline is approaching in **1 day**, could you please have a look at our response? We are looking forward to your reply!
>
> Feel free to let us know if you have any other concerns. Thanks for your time and effort!
>
> Best Regards,
>
> Authors of Submission 19827

---

> > ### Comment · Area_Chair_qDXX · 2025-08-06
> > **Request to Respond to Author Rebuttal and Participate in Discussion**
> >
> > Dear Reviewer  mtFH,
> >
> > As the Area Chair for this paper, I would like to kindly request your active participation in the ongoing discussion phase—particularly as the paper has received mixed borderline reviews, and the authors have requested your feedback on the rebuttal multiple times.  Additionally, I would like to clarify the requirements related to the Mandatory Acknowledgment process. To fulfill this requirement, reviewers are expected to:
> >
> > (i) Carefully read the author rebuttal,
> >
> > (ii) Engage in meaningful discussion with the authors—and preferably also with fellow reviewers.
> >
> > (iii) Ask questions, consider responses, and actively participate in the exchange,
> >
> > (iv) Clearly articulate any unresolved concerns to give authors a fair opportunity to respond. Please avoid situations where the discussion implies “everything is great,” but the final justification form states otherwise. The discussion phase is designed to surface and clarify such issues.
> >
> > Kindly note that clicking the “Mandatory Acknowledgment” checkbox prematurely does not exempt reviewers from participating in the discussion. Reviewers who do not contribute meaningfully may be flagged using the “Insufficient Review” button, in line with this year’s responsible reviewing guidelines.
> >
> > Thank you for your time and thoughtful contributions to the review process.

---

### Official Review · Reviewer_78xg · 2025-07-02

**Clarity:** 3
**Significance:** 3
**Originality:** 3
**Rating:** 5
**Confidence:** 3

**Summary:**

This paper introduces HoloLLM, a multimodal large language model (MLLM) designed to integrate a range of sensing modalities beyond vision, for typical home environments. One contribution is UMIP, a framework that unifies pre-aligned modalities with text modality. The authors also propose a human–VLM collaborative pipeline and use it to construct two new benchmarks that facilitate language grounded human sensing. Experimental results on these benchmarks demonstrate that HoloLLM outperforms previous state-of-the-art models.

**Questions:**

1) see weaknesses
2) what message does Figure 6 right column intend to convey?

**Ethical Concerns:**

["NO or VERY MINOR ethics concerns only"]

**Final Justification:**

The rebuttal clarified my questions. The paper makes a nice contribution to an under-explored area (i.e., unconventional modalities).

**Limitations:**

yes

**Paper Formatting Concerns:**

no concern

**Quality:**

3

**Strengths And Weaknesses:**

Strengths.

- Exploring MLLMs that extend beyond vision to incorporate additional modalities available from embodied agents (e.g., robots) is an important yet underexplored area.

- This paper proposes UMIP, a simple yet effective architecture that integrates modality-specific encoders to address challenges in data-scarcity modality-to-text alignment.

-  To support evaluation, the authors introduce two new benchmarks for language-grounded human sensing, built by extending existing datasets.

Weaknesses

-	What are the key architectural or methodological differences between HoloLLM and the baseline approaches it was compared against? HoloLLM shows significantly improved performance, often by large margins. It would be helpful to clarify whether this improvement is primarily due to its architectural design (e.g., UMIP) or differences in training data. Specifically, were the modality-specific encoders in the baseline models trained using the same data as HoloLLM, or were there discrepancies in training conditions that could account for the performance gap?

-	In Table 1 and Table 3, all different modalities are reported separately. What is the performance by integrating these modalities together?

-	What is the inference time latency?

---

> ### Author Rebuttal · Authors · 2025-07-31
>
> We sincerely appreciate the reviewer 78xg for the insightful questions and detailed review. Here, we answer all the questions and add extensive experiments for clarification.
>
> **Q1: What are the key architectural or methodological differences between HoloLLM and the baseline approaches it was compared against? It would be helpful to clarify whether this improvement is primarily due to its architectural design (e.g., UMIP) or differences in training data. Specifically, were the modality-specific encoders in the baseline models trained using the same data as HoloLLM, or were there discrepancies in training conditions that could account for the performance gap?**
>
> **Answer:** We appreciate the suggestion to clarify the key differences between HoloLLM and the two baseline approaches we adopted in the ablation study (Tab.3 in the original manuscript). Here, we summarize the key differences among the three models.
>
> |Models|Encoders|Projectors|Training Data|
> |-|-|-|-|
> |Baseline|The unified CLIP encoder $E_{CLIP}(\cdot)$|Q-former projector with learnable queries and CLIP features as keys and values.|The same training data (detailed in Appendix B.2) is used for both pretraining the tailored modality-specific encoders and fine-tuning the projectors. There are no differences in training data between HoloLLM and the other two baseline models.|
> |Baseline+Tailored Encoder|The tailored modality-specific encoder $E_T^m(\cdot)$| Q-former projector with learnable queries and modality-specific features as keys and values.|Please see above.|
> |Baseline+Tailored Encoder+UMIP (HoloLLM)| Both the unified CLIP encoder $E_{CLIP}(\cdot)$ and the tailored modality-specific encoder $E_T^m(\cdot)$|UMIP with CLIP features as coarse queries and modality-specific features as keys and values.|Please see above.|
>
> In summary, we ensure a fair comparison, and the difference between HoloLLM and the baseline models lies entirely in our architectural designs, including the tailored modality-specific encoders and the UMIP. Specifically, the tailored encoders effectively capture heterogeneous modality-specific features within sensing modalities through pretraining. Meanwhile, the UMIP achieves efficient multimodal alignment while preserving modality-specific discriminative information. Both components play critical roles in significantly improving performance, as demonstrated by the ablation.
>
> We will incorporate the clarification regarding the differences between HoloLLM and the other baseline models in the revised manuscript.
>
> **Q2: In Table 1 and Table 3, all different modalities are reported separately. What is the performance of integrating these modalities together?**
>
> **Answer:** We appreciate the comment regarding the multimodal fusion capabilities of HoloLLM. We have provided experimental results of naïve multimodal fusion, simply concatenating all multimodal tokens, in Appendix C.3 in the manuscript. The results show that naïve multimodal fusion can improve action recognition performance. However, for action QA and captioning tasks, naïve multimodal fusion fails to improve performance. We conjecture that this is because naïve multimodal fusion does not fully leverage complementary information across modalities nor effectively reduce redundancy.
>
> To make a stronger multimodal fusion baseline, we further supplement an experiment to explore stronger adaptive fusion techniques, including weighted sum (WS) and max pooling (MP) across multimodal tokens.
>
> ||MM-Fi||||XRF55|||
> |-|-|-|-|-|-|-|-|
> | |HAR|QA|Caption| |HAR|QA|Caption|
> |V|80.6|79.5|25.7|V|28.9|25.9|19.8|
> |M|61.0|61.4|24.5|I|28.3|22.1|17.1|
> |V+M (Naive)|**84.6**|66.4|25.0|V+I (Naive)|**34.3**|21.5|19.2|
> |V+M (WS)|82.6|**81.38**|**26.9**|V+I (WS)|30.91|**26.24**|**20.6**|
> |V+M (MP)|80.6|80.43|26.7|V+I (MP)|28.9|20.48|18.9|
>
> The results show that the weighted sum fusion techniques outperform single-modality and naïve multimodal fusion on Action QA and captioning tasks. We believe that more advanced, learning-based multimodal fusion techniques could further enhance performance. We hope that HoloLLM can serve as a challenging benchmark to inspire the research community toward more insightful contributions in this multimodal fusion technique.
>
> **Q3: What is the inference time latency?**
>
> **Answer:** We appreciate the suggestion to show inference latency, which is critical for real-world applications. Specifically, we select the video and WiFi as representative common and sensing modalities to conduct a fair comparison on the same GPU setting, i.e., one A100. We report the inference times for ImageBind (encoder-based method), OneLLM (projector-based method), and HoloLLM on the MM-Fi dataset across all three tasks under 'random' settings. Inference latency is calculated by averaging the time across all batches with a test batch size of 16. The inference latencies of HoloLLM and other state-of-the-art MLLMs are summarized in the following table.
>
> |Models| | | |Inference latency (seconds/batch)| | |
> |-|-|-|-|-|-|-|
> | |HAR| |Action QA| |Action Caption| |
> | |V|W|V|W|V|W|
> |ImageBind|21.1|15.2|23.3|18.0|31.0|27.4|
> |OneLLM|20.0|15.1|23.0|16.5|30.0|24.1|
> |HoloLLM|19.7|14.7|22.0|16.1|29.5|22.5|
>
> The inference latencies of all MLLMs are nearly identical. We think this is reasonable since the main inference bottleneck occurs on the LLM side, and all models use the same LLaMA2-7B backbone.
>
> **Q4: What message does Figure 6 right column intend to convey?**
>
> **Answer:**  The message conveyed by the right column of Fig. 6 in the manuscript highlights the significance of our UMIP.
>
> As discussed in **Q1**, HoloLLM w/o UMIP (i.e., Baseline + Tailored Encoder) uses the original Q-former as projector, whereas HoloLLM employs the proposed UMIP as projector. The right column of Fig. 6 shows that the multimodal tokens generated by UMIP in HoloLLM, along with text tokens from ground-truth captions, are correctly grouped by action category. Consequently, UMIP facilitates better multimodal alignment with the LLM compared to the original Q-former. The right column of Fig. 6 also provides an intuitive explanation for the performance improvement brought by UMIP on tasks that require a deeper understanding of language-based instructions and action categories, such as Action QA in Tab. 3.
>
> We really appreciate the reviewer's detailed comments on our figure, which is truly a bit vague in this aspect. We will modify Section 4.4 for better clarification in the revised manuscript.

---

> ### Author Response · Authors · 2025-08-04
> **Looking forward to your reply**
>
> Dear Reviewer 78xg,
>
> As the discussion phase deadline is approaching in **2 days**, could you please have a look at our response? We are looking forward to your reply!
>
> Feel free to let us know if you have any other concerns. Thanks for your time and effort!
>
> Best Regards,
>
> Authors of Submission 19827

---

> ### Author Response · Authors · 2025-08-05
> **Looking forward to your reply**
>
> Dear Reviewer 78xg,
>
> As the discussion phase deadline is approaching in **1 day**, could you please have a look at our response? We are looking forward to your reply!
>
> Feel free to let us know if you have any other concerns. Thanks for your time and effort!
>
> Best Regards,
>
> Authors of Submission 19827

---

> > ### Comment · Reviewer_78xg · 2025-08-05
> >
> > Thank you for your responses to my questions. I appreciate the detailed comparisons and remain positive about this work.

---

### Official Review · Reviewer_DvFs · 2025-07-03

**Clarity:** 3
**Significance:** 2
**Originality:** 2
**Rating:** 4
**Confidence:** 3

**Summary:**

This paper presents HoloLLM, a large language model that understands human actions using not just visual data, but also other sensor types like LiDAR, infrared, WiFi, and mmWave radar. These sensors work better than cameras in dark, blocked, or privacy-sensitive places.

Since these sensor types don’t have much paired text data and are very different from images, the authors design two solutions: (1) Tailored encoders to extract useful features for each sensor type. (2) UMIP, a module that combines those features with CLIP embeddings and gradually aligns them to the language model.

They also build a new dataset with human + AI collaboration for tasks like action recognition, question answering, and captioning. Experiments show that HoloLLM works much better than previous models (up to 30% better) on these tasks.

**Questions:**

1. Your human-VLM data curation pipeline is insteresting, but how do you ensure annotation quality—especially for captions generated by LLaVA-Video? Could you report: Inter-annotator agreement (if any human review was done); Human evaluation of generated captions (e.g., fluency, accuracy scores); Percentage of samples verified or discarded. Addressing this would increase trust in your benchmarks, which are key to evaluating your model. Without quality control, the evaluation may be biased or noisy.

2. HoloLLM is framed as a general multisensory foundation model, yet it is only evaluated on MM-Fi and XRF55, which have similar indoor settings and overlapping sensor types. How well does HoloLLM generalize to new domains, sensors, or unseen environments? If feasible, please include zero-shot or few-shot results on other datasets, such as SenseFi or Widar.

**Ethical Concerns:**

["NO or VERY MINOR ethics concerns only"]

**Final Justification:**

The authors have addressed my main concerns with new experiments on annotation quality, adaptive fusion, and generalization to new datasets and modalities.While cross-environment performance is still limited and the novelty is moderate, I maintain my Borderline accept recommendation.

**Limitations:**

The authors include a brief limitations section noting that HoloLLM is currently limited to human action understanding tasks (recognition, QA, captioning), and does not yet handle higher-level capabilities like task planning or real-world agent control. This is a good start.  Technical limitations are not fully addressed because the model is only evaluated on two similar indoor datasets.

**Paper Formatting Concerns:**

There are no critical formatting violations.

**Quality:**

3

**Strengths And Weaknesses:**

Quality: (1) Strengths: The technical methodology is sound and well-motivated. The UMIP design is carefully engineered to address the unique challenges of data scarcity and sensor heterogeneity. The experiments are broad, covering five sensing modalities across two datasets and three settings (Random, CrossSub, CrossEnv). Ablation studies are helpful in validating both the tailored encoders and UMIP. (2) Weaknesses: The empirical rigor could be improved: Some accuracy (e.g., CrossEnv) are small and need stronger design support. The validation of the human-VLM collaborative pipeline is weak—there is no measure of annotation quality or consistency. No comparison with multimodal transformers trained end-to-end or with stronger adaptive fusion techniques beyond naive fusion.

Clarity: (1) Strengths: The paper is logically organized and generally easy to follow at a high level. Motivation is concrete (smart homes, privacy, occlusion), and the contribution is clearly stated. (2) Weaknesses: Key sections, especially Section 3.2 (UMIP), are too dense, please consider adding pseudocode or a simplified diagram of the iterative process. The link between CLIP embeddings and non-visual modality alignment needs clearer explanation.

Significance: (1) Strengths: The paper addresses an important and under-explored problem: language grounding with non-visual sensing. (2) Weaknesses: Broader generalization across datasets or unseen modalities is not demonstrated.

Originality: (1) Strengths:The two-stage training strategy and the human-VLM collaborative annotation pipeline are thoughtful and well-justified. (2) Weaknesses: The idea of cross-attention-based fusion is not new—more emphasis is needed on what differentiates UMIP from prior Q-former-style projectors (e.g., OneLLM, BLIP-2). (2) Need to better position UMIP versus recent work on universal modality encoders (e.g., ImageBind, OmniFusion, UnifiedIO). (3) The use of CLIP for non-visual modality embedding needs stronger justification, or at least empirical support.

---

> ### Author Rebuttal · Authors · 2025-07-31
>
> We appreciate the reviewer DvFs for the constructive comments and the appreciation. Here we answer all the questions by extensive experiments and clarification.
>
> **Q1: Some accuracy (CrossEnv) is small and need stronger design.**
>
> **Answer:** We appreciate the suggestion and admit the issue. In fact, the “cross-subject” and “cross-env” generalization of some modalities (e.g., WiFi) is very challenging, as theoretically they are susceptible to variations in subjects and environments, and the data scarcity makes these tasks even harder. Though the tasks are hard, HoloLLM still outperforms other SOTA MLLMs in both settings by a large margin. In future work, we aim to enhance the generalization ability of HoloLLM by constructing a large-scale sensing dataset. Specifically, we plan to integrate existing public datasets [1, 2] and collect additional data from a broader range of environments (e.g., outdoor scenarios) and tasks (e.g., task planning and agent action generation). By fine-tuning HoloLLM with this new dataset, we believe that HoloLLMv2 will obtain strong generalization ability. This work focuses on aligning new yet powerful sensing modalities with LLM, given limited data and various modalities, and thus offers a novel technical solution with a comprehensive benchmark.
>
> **Q2: There is no validation of the human-VLM collaborative pipeline.**
>
> **Answer:** As suggested, we invite 10 volunteers and sample 1% caption data from MM-Fi (165 / 16,448) and XRF55 (198 / 19,800) for inter-annotator agreement evaluation (Fleiss’ Kappa score [6]). Specifically, the volunteers are asked to perform a three-class classification task: “Does the caption match (label 2), partially match (label 1), or not match (label 0) the video?” The matching score for each class is defined as: # of samples with label n / # of all samples. Results are summarized below:
>
> | |MM-Fi|XRF55|
> |-|-|-|
> |Matching scores (match / partially match / not match) |91.21% / 8.79% / 0%|85.76% / 14.24% / 0%|
> |Fleiss Kappa score|0.7354|0.6278|
>
> The results indicate that annotations generated by our pipeline closely match the human action videos, with no samples labeled as ‘not match’ within the sampled data. Additionally, the two volunteers achieve substantial agreement, with Fleiss’ Kappa scores > 0.6 on both datasets. Furthermore, we will release all annotations along with the HoloLLM source code.
>
> **Q3: Comparison with methods trained end-to-end or with stronger adaptive fusion techniques.**
>
> **Answer:** We have already compared HoloLLM with end-to-end trained SOTA methods, including ImageBind and OneLLM. Specifically, ImageBind is trained end-to-end using an InfoNCE loss across modalities, while OneLLM is fine-tuned end-to-end with a task-specific instruction tuning loss. The results are presented in Tab. 1 and Tab. 2 in the original manuscript.
>
> In addition, we agree with the suggestion that adopting naïve fusion (concatenating all multimodal tokens) alone is insufficient. Therefore, we add experiments to explore more advanced adaptive fusion techniques, including weighted sum (WS) and max pooling (MP) across multimodal tokens.
>
> ||MM-Fi||||XRF55|||
> |-|-|-|-|-|-|-|-|
> | |HAR|QA|Caption| |HAR|QA|Caption|
> |V|80.6|79.5|25.7|V|28.9|25.9|19.8|
> |M|61.0|61.4|24.5|I|28.3|22.1|17.1|
> |V+M (Naive)|**84.6**|66.4|25.0|V+I (Naive)|**34.3**|21.5|19.2|
> |V+M (WS)|82.6|**81.38**|**26.9**|V+I (WS)|30.91|**26.24**|**20.6**|
> |V+M (MP)|80.6|80.43|26.7|V+I (MP)|28.9|20.48|18.9|
>
> The results show that the weighted sum fusion technique outperforms single-modality and naïve multimodal fusion on Action QA and captioning tasks. We believe that more advanced, learning-based multimodal fusion techniques could further enhance performance. We hope that HoloLLM can serve as a challenging benchmark to inspire the research community toward more insightful contributions in multimodal fusion techniques.
>
> **Q4: Clarify Section 3.2 (UMIP) by adding pseudocode and the link between CLIP embeddings and multimodal alignment.**
>
> **Answer:** Due to the limited characters, we will **directly add** the pseudocode in the revised manuscript. Here, we further clarify the link between CLIP embeddings and non-visual modality alignment. As illustrated in Eq.(1), we leverage CLIP to generate pre-aligned embeddings $Y_{CLIP}^m$  for modality $m$, which are condensed using adaptive average pooling and used as queries (Eq.(3)). **Therefore, CLIP provides pre-aligned queries that contain coarse modality-specific features.** This enables the UMIP to adaptively identify aligned modality-specific features via cross-attention and progressively integrate them for enhancement (Eq.(4)).
>
> **Q5: Demonstrate generalization of HoloLLM to new domains (e.g., SenseFi), sensors, or unseen environments?**
>
> **Answer:** As suggested, we add experiments for new domains and sensors.
>
> (1) Generalization to New Datasets.
>
> We report zero-shot results on SenseFi [8]. Specifically, SenseFi performs human sensing using WiFi-CSI signals and includes six novel action categories. We then apply the ‘Baseline’ model (as detailed in line 248) and HoloLLM trained on the MM-Fi dataset under the ‘random’ setting to perform zero-shot human action QA on the official SenseFi test set, which contains 264 samples.
>
> |Model|Action QA|
> |-|-|
> |Baseline|5.43|
> |HoloLLM|16.67|
>
> In zero-shot settings, HoloLLM’s superior performance shows its strong generalization ability and discriminability.
>
> (2) Generalization to Unseen Sensing Modalities
>
> HoloLLM can integrate new modalities more efficiently than other MLLMs, owing to the novel architecture of UMIP. Specifically, we extensively pretrain a lightweight, modality-specific tailored encoder for each new modality to sufficiently capture fine-grained features. The new modality is then efficiently integrated into HoloLLM by UMIP with minor data and fine-tuning.
>
> We add an experiment to demonstrate the superior data efficiency of HoloLLM compared to the ‘Baseline’ model on two novel modalities, audio [9] and UWB [10]. The results are summarized below.
>
> | |HAR| |Action QA| |
> |-|:-:|:-:|:-:|:-:|
> | |Audio|UWB|Audio|UWB|
> |Baseline|37.71|9.46|30.73|11.33|
> |HoloLLM|63.94|16.77|54.17|16.45|
>
> Compared to the ‘Baseline’ model, HoloLLM can be efficiently generalized to novel modalities with minor data and fine-tuning.
>
> (3) Generalization to Unseen Environments.
>
> We already presented the results in Tab. 1 and Tab. 2 in the original manuscript under the ‘CrossEnv’ setting.
>
> **Q6: Position UMIP versus prior Q-former-style projectors and universal modality encoders.**
>
> **Answer:** The differences between HoloLLM and existing methods can be categorized into four key aspects. **(1) Task.** It deals with a more challenging problem on data-scarce yet powerful sensing modalities with heterogeneous features. **(2) Model Architecture.** It proposes a novel projector that incorporates fine-grained, modality-specific features for enhancing multimodal alignment. **(3) Training Strategy.** It presents a two-stage training strategy to sufficiently capture the modality-specific features and efficiently align multimodal inputs with the text space. **(4) Benchmark Results.** It establishes the first multisensory benchmark for human perception and reasoning.
>
> We also present a detailed comparison of HoloLLM with Q-former-style projectors (OneLLM [7], BLIP-2 [11]) and universal modality encoders (ImageBind [12], OmniFusion [13], UnifiedIO [14]). Due to the limited characters, we only include OmniFusion and UnifiedIO here. Please refer to our response to **Reviewer 4VBc Q1** for all other methods.
>
> | |Task|Model Architecture|Training Strategy|Benchmark Results|
> |-|-|-|-|-|
> |HoloLLM (ours)|Align **data-scarce** but powerful **sensing modalities** (e.g., WiFi) with LLM using only limited data, while preserving their **heterogeneous features** for seamless human perception and reasoning.|A unified multimodal projector that considers **modality-specific features** by taking features from CLIP and **tailored encoders** as coarse queries and keys/values.|**Two-stage training.** (1) Pretrain tailored encoders to capture modality-specific features. (2) Fine-tune the UMIP to generate discriminative multimodal tokens aligned with text.|HoloLLM achieves SOTA performances on two **human sensing** datasets, establishing the first **multisensory benchmark** for human perception and reasoning.|
> |OmniFusion [13]|Fuse tangent images, RGB images, and 3D geometric features to **generate high-quality dense depth maps.**|A typical UNet architecture to generate depth map.|Train the **whole UNet** from scratch with the task-specific loss for depth image prediction.|Evaluate on depth image generation benchmarks.|
> |UnifiedIO [14]|Align vision-related modalities (e.g., images, bounding boxes) with text for sparse and dense vision tasks (e.g., VQA or segmentation).|Pure transformer encoder-decoder architecture.|Train the whole transformer from scratch via **large-scale vision-text pairs**.|Only evaluated on vision-language benchmarks for sparse and dense visual tasks, such as GRIT.|
>
> **Q7: Justify the use of CLIP for non-visual modality embedding.**
> **Answer:** Our use of CLIP for non-visual modality embeddings is inspired by both OneLLM [7] and Meta-Transformer [4], which suggest that **a well-pretrained transformer can serve as a universal cross-modal encoder.**
> Consequently, HoloLLM adopts the CLIP vision encoder as a unified encoder to generate pre-aligned initial embeddings for each modality.
>
> However, we find that the CLIP encoder is challenging to "learn heterogeneous characteristics of rare sensing modalities", thus we design tailored encoders to capture modality-specific features and integrate them via UMIP, denoted as one of the main novelties. As illustrated in Tab.3, the modality-specific knowledge is critical to enhancing the discriminability.
>
> **References**
>
> Due to limited characters, please refer to the responses of Reviewer 4VBc for references.

---

> > ### Comment · Reviewer_DvFs · 2025-08-08
> >
> > Thank you for addressing my comments and adding the new results on the human–VLM collaborative pipeline, adaptive fusion, and the explanation on generalization. I agree that cross-environment adaptation is challenging. I am still leaning toward the positive side, so I will keep my score unchanged.

---

> ### Author Response · Authors · 2025-08-04
> **Looking forward to your reply**
>
> Dear Reviewer DvFs,
>
> As the discussion phase deadline is approaching in **2 days**, could you please have a look at our response? We are looking forward to your reply!
>
> Feel free to let us know if you have any other concerns. Thanks for your time and effort!
>
> Best Regards,
>
> Authors of Submission 19827

---

> ### Author Response · Authors · 2025-08-05
> **Looking forward to your reply**
>
> Dear Reviewer DvFs,
>
> As the discussion phase deadline is approaching in **1 day**, could you please have a look at our response? We are looking forward to your reply!
>
> Feel free to let us know if you have any other concerns. Thanks for your time and effort!
>
> Best Regards,
>
> Authors of Submission 19827

---

### Author Response · Authors · 2025-08-09
**Summary of the key improvements during the rebuttal phrase**

Dear reviewers,

We would like to express our heartfelt gratitude for the reviewers' valuable time, expertise, and thoughtful feedback on our manuscript. The insightful comments have greatly contributed to improving the quality and rigor of our work.

We are encouraged that all reviewers acknowledged the novelty and significance of our contribution:

- "The paper addresses an important and under-explored problem." (Reviewer DvFs)

- "This paper proposes UMIP, a simple yet effective architecture." (Reviewer 78xg)

- "Reasonable experiments demonstrate the effectiveness of the proposed method." (Reviewer mtFH)

- "The motivation is clearly stated, and the paper is well organized." (Reviewer 4VBc)

At the same time, we recognize that several issues require further clarification or substantiation. In response, we have carefully addressed all reviewer concerns and conducted additional experiments to strengthen our claims. Key improvements include:

1. **Expanded Multimodal Fusion Techniques.**  Included two more advanced multimodal fusion techniques (Max Pooling, Weighted Sum) to demonstrate the superior multimodal fusion capabilities of HoloLLM.

2. **Two New Experiments on Generalization of HoloLLM.**  Conducted zero-shot Action QA on the SenseFi dataset and evaluated generalization to two novel modalities (Audio, UWB), demonstrating the superior generalization ability of HoloLLM.

3. **New Evaluation on LanguageBind.**  Added the performance of a more recent unified multimodal encoder, LanguageBind, on both MM-Fi and XRF55 datasets in Tab. 1 and Tab. 2, establishing a more comprehensive multisensory human sensing and reasoning benchmark.

4. **Inference Time Latency.** Added inference latency of HoloLLM and other SOTA MLLMs across all three tasks for more comprehensive comparisons.

5. **Human-VLM Collaborative Pipeline Validation.** Invited 10 volunteers and conducted an inter-annotator agreement evaluation, showing high quality and consistency of the annotations generated by our data curation pipeline.

6. **Motivation Emphasis.** Discussed differences between HoloLLM and other SOTA MLLMs (Diverse Projectors, Universal Modality Encoders) regarding target tasks, model architecture, training strategy, and benchmark results, highlighting the significance of HoloLLM.

7. **Writing Clarity Improvement.** Added pseudocode of the iterative process within UMIP and positioned the HoloLLM versus the other two baseline models in the ablation study for clearer explanations.

Our HoloLLM makes significant contributions by **introducing a novel task, establishing a pioneering benchmark with curated datasets, and proposing a simple yet effective method.** During the rebuttal phase, we made extensive additions and revisions to further strengthen and enrich the manuscript in terms of experiments, analyses, and writing quality. We believe the refined HoloLLM can **serve as a comprehensive benchmark to inspire the research community toward more insightful contributions in this multimodal LLM domain.**

Best Regards,

Authors of Submission 19827

---

### Note · Authors · 2025-08-14

Dear reviewers,

Once again, we sincerely thank all reviewers for your efforts and constructive feedback, which greatly improved our work.

We are encouraged that all reviewers appreciated the novelty and significance of our contribution:


>|Reviewer|Comments|
>|-|-|
>|DvFs|"The paper addresses an important and under-explored problem."|
>|78xg|"This paper proposes UMIP, a simple yet effective architecture."|
>|mtFH|"Reasonable experiments demonstrate the effectiveness of the proposed method."|
>|4VBc|"The motivation is clearly stated, and the paper is well organized."|

At the same time, we recognize that several issues require further clarification or substantiation. In the rebuttal, we substantially strengthened the work via (1) extensive empirical studies on multimodal fusion techniques, generalization of HoloLLM, more recent SOTA methods, inference latency, and our data curation pipeline; (2) demonstrating significant differences of HoloLLM with other SOTA MLLMs regarding the target task, model architecture, training strategy, and benchmarks result; and (3) improving the motivation elaboration and writing clarity. We invite all reviewers to refer to our official comment on the “[summary of the rebuttal](https://openreview.net/forum?id=cHMP2IAhML&noteId=UlsQjXNo6Z)” for more details.

We believe that these improvements make this work stronger, encompassing **a novel task, a comprehensive benchmark with curated datasets and SOTA M-LLMs, and a simple yet effective method as a baseline**. We believe HoloLLM could inspire the research community, further enhancing the research on multimodal LLM for data-scarce but useful modalities.

Best Regards,

Authors of Submission 19827

---

### Decision · Program_Chairs · 2025-09-17

**Decision:**

Accept (poster)

**Comment:**

Summary:
The paper presents HoloLLM, an MLLM that integrates rare sensing modalities to enable seamless human perception and reasoning across heterogeneous real-world scenarios. To efficiently align sensing modalities with text using minimal fine-tuning, the paper proposes the Universal Modality Injection Projector (UMIP) along with modality-specific discriminative features.

Key strengths highlighted by the reviewers:
(i) The paper addresses an important and under-explored problem (i.e., unconventional modalities).
(ii) To formulate the new task, the authors curated new datasets and proposed a comprehensive benchmark, which will hopefully support future research.
(iii) Experiments demonstrate the effectiveness of the proposed method.
(iv) The motivation is clearly stated, and the paper is well organized.
(v) The proposed UMIP is a simple yet effective architecture.

Key weaknesses remaining after the rebuttal:
(i) The model’s cross-environment performance remains limited.
(ii) Novelty is moderate.
(iii) The distinction between the proposed contributions and existing approaches such as LanguageBind needs to be clarified in the final draft.
(iv) The paper primarily focuses on aligning a single modality to text; additional experiments on multi-modal fusion (as presented in the rebuttal) should be included in the final draft.

Decision summary:
The paper received one Accept, two Borderline Accepts, and one Borderline Reject. The authors provided additional results and explanations in the rebuttal, and reviewers agreed that most of their concerns were addressed. Although the architectural novelty of the method is limited, the area chair believes that the paper tackles an important and under-explored problem. The proposed dataset and models are expected to be a valuable contribution to the community; hence, the recommendation is to accept the paper. The authors are expected to address the remaining weaknesses in the final draft, include additional explanations and results reported in the rebuttal to the final draft or appendix,  and open-source their code and data as stated in the original submission.